
# Process-based flood damage modelling relying on expert knowledge: a methodological contribution applied to agricultural sector

Pauline Brémond[1], Anne-Laurence Agenais[1], Frédéric Grelot[1], and Claire Richert[2]

[1]INRAE, G-EAU, Univ Montpellier, AgroParisTech, BRGM, CIRAD, IRD, INRAE, Institut Agro, Montpellier, France
[2]ITK, Clapiers, France

**Correspondence:** P.Brémond
(pauline.bremond@inrae.fr)

**Abstract.** Flood damage assessment is crucial for evaluating flood management policies. In particular, properly assessing damage to the agricultural assets is important because they may have greater exposure and are complex economic systems. The modelling approaches used to assess flood damage are of several types and can be fed by damage data collected post-flood, from experiments or based on expert knowledge. The process-based models fed by expert knowledge are subject of research and also widely used in an operational way. Although identified as potentially transferable, they are in reality often case-specific and difficult to reuse in time (updatbililty) and space (transferability). In this paper, we argue that process-based models are not doomed to be context specific as far as the modelling process is rigorous. We propose a methodological framework aiming at verifying the conditions necessary to develop these models in a spirit of capitalisation by relying on four axes which are: i/ the explicitation of assumptions, ii/ the validation, iii/ the updatability, iv/ the transferability. The methodological framework is then applied to the model we have developed in France to produce national damage functions for the agricultural sector. We show in this paper that the proposed methodological framework allows an explicit description of the modelling assumptions and data used, which is necessary to consider a reuse in time or a transfer to another geographical area. We also highlight that despite the lack of feedback data on post-flood damages, the proposed methodological framework is a solid basis to consider the validation, transfer, comparison and capitalisation of data collected around process-based models relying on expert knowledge. In conclusion, we identify research tracks to be implemented to pursue this improvement in a spirit of capitalisation and international cooperation.

## 1 Introduction

Worldwide, flooding generate huge damage (van Loenhout et al., 2020) estimated at 58 billion EUR (75 billion USD) per year (Alfieri et al., 2017). The EU Floods Directive (Directive 2007/60/EC) requires Member States first, to map flood extent and assets at risk; second to coordinate measures to reduce this flood risk. Every Member States are confronted to this challenge to decrease total flood damage while urban assets keep on developing in flood prone





area (Rojas et al., 2013). To face this challenge, flood management usually mix several types of approach at river
basin level. Agricultural areas are globally generate less damage than urban ones (1% only of the total damage in
Europe (Alfieri et al., 2017)). As a consequence, protection measures such as dykes are usually dedicated to protect
urban area. Farmers are rather seen as potential contributors to reduce flood risk either by changing their practices
(O'Connell et al., 2007; Posthumus and Morris, 2010) or by using agricultural lands to give more room for water
flooding which involves increasing their exposure (Morris et al., 2010). However, the second type of measures raise
many questions on acceptability and compensations (Zandersen et al., 2020; Erdlenbruch et al., 2009; Posthumus
et al., 2008, 2010). Then properly evaluate flood damage on agriculture becomes a real issue for two main reasons.
First, evaluating flood damage on agriculture is necessary to justify the efficiency of the policy and then the choice
that can be done between several options. This is usually done by performing Cost Benefit Analysis which requires
developping flood damage functions (Jonkman et al., 2008; Merz et al., 2010) Second, even if the project is efficient,
the acceptability of those measures requires involving farmers (Posthumus et al., 2008) and introducing compensation
payments (Erdlenbruch et al., 2009). To reach this goal, developing a comprehensive model to evaluate flood damage
on farms is necessary. In particular, to discuss and build a trusting relationship with farmers that may be over
exposed, this model needs to reflect as much as possible what happens to them in case of flooding.

Several classifications of the methods used to model flood damage can be found in the litterature (Jongman
et al., 2012; Davis and Skaggs, 1992; Merz et al., 2010; Molinari et al., 2020; Malgwi et al., 2021). However, these
classifications are not operative because they mix the modelling methods and the data necessary to feed the models.
Presenting the modelling methods separately from the data needed to feed them provides greater clarity. The
strategies generally adopted to model flood damages are: (i) data driven modelling, (ii) conceptual modelling, (iii)
process-based modelling. To feed these models, different types of data can be used: (i) damage observation data,
(ii) data from expert knowledge, (iii) data from experiments. Data driven modelling approaches requires damage
observation data. Conceptual modelling are more often used to evaluate indirect damage with input-output (IO)
models (Hallegatte, 2008; Van der Veen et al., 2003; Hallegatte, 2014; Crawford-Brown et al., 2013; Xie et al.,
2012) or computable general equilibrium (CGE) models (Xie et al., 2014; Rose and Liao, 2005; OCDE, 2014).
They are appropriate for indirect and large scale damage evaluation but not for sectoral damage evaluation at
micro and meso scales. Process-based modelling can be fed by expert knowledge or experimental data. Experiments
require very significant monetary and time investments. Most often process-based modelling approaches are fed with
expert knowledge. It is recommended to have experienced interviewers, who also have some knowledge of making
damage estimates (Davis and Skaggs, 1992). To illustrate these categories of modelling approaches, let us take
the example of flood damage assessment models developed in Germany and the United Kingdom. In Germany, a
huge effort to collect ex post damage data has been carried out (Thieken et al., 2017) and the models developped
for residential (FLEMO-ps) (Thieken et al., 2008a) and economic assets (FLEMO-c) (Kreibich et al., 2010) are
data driven models. On the contrary, the flood damage functions that have been etablished in United Kingdom
by the Flood Hazard Research Center are process-based models fed with expert knowledge (Penning-Rowsell and



Chatterton, 1977; Penning-Rowsell et al., 1992, 2005, 2013; Priest et al., 2021b). The flood damage models INSYDE (Dottori et al., 2016) in Italy or **floodam** (Grelot and Richert, 2019) in France are also part of this category. Each of these methods has its advantages and drawbacks. For data-based approaches, it remains difficult to systematically collect individual data on a large scale. For process-based approaches, the understanding of processes often remains too incomplete (Merz et al., 2010; Meyer et al., 2012, 2013). Moreover, process-based modelling approaches are often pointed out as being context specific and not allowing capitalisation of modelling efforts to other contexts.

Flood damage on economic activites such as farms is classically estimated by the loss of added value (Penning-Roswell et al., 2005; Brémond and Grelot, 2010). The loss of added value corresponds to the decrease in product minus the variation in production costs due to flooding (Brémond et al., 2013). Due to flood impacts, the farmer will make some choices which will lead to variation in production costs. Some may be saved while others may increase (treatment, tillage, for instance). In clear, assessing flood damage on farms requires modelling both the biophysical damage processes to determine the damage levels of the components and the behavior of farmers to determine the variations in production costs. However, a litterature review conducted by Brémond et al. (2013) on flood damage modelling for agricultural activities showed that many simplifications are usually done. Although several studies (42) have been carried out at international level, no method was directly transferable to evaluate agricultural damage at national scale in France. In particular, the key points were that:

- few methods considered farm as an economic activity and only considered the loss of yield;

- the biophysical processes considered were not explicit

- the loss of yield is estimated in function of period of the year but not in function of the vegetative cycle which hinder the transferability to other geographical context;

- the implications of flooding on farmers' actions were not explicitly considered and the variation of charges were not transferable;

- the implications of flooding for perenial crops were not taken into account;

- no example of validation of modelling assumptions were found in the litterature.

Since 2013, based on Agenais et al. (2013), Molinari et al. (2019b) and Scorzini et al. (2020) implemented a flood damage model to crops but despite the efforts made, the way in which the experts' knowledge was collected and formalized is not made explicit, particularly with regard to the assumptions made about the processes and behaviors of the farmers actually taken into account.

No data diven models for agricultural sector was find in the litterature. In germany, no model such as FLEMOps or FLEMOc exists for agriculture (Thieken et al., 2008b). To evaluate agricultural damage in the MEDIS project, Forster et al. (2008) extrapolated yield loss estimation based on one specific flood in Germany. This can be explained by the fact that little sinsitrality data is available for the agricultural sector. The private insurance for flood crop





losses is low (Priest et al., 2021a; Browne, 2000) and no private insurance for overall agricultural damage exist as for example for soil erosion. Conceptual models are not suitable for assessing damage at the watershed or farm level (Meyer et al., 2013). As Brémond et al. (2013) state the assessment of agricultural damage requires a fine-grained understanding of the types of damage to be considered in addition to crop loss alone. Damage to agricultural assets results both from complex biophysical processes and from repair and recovery actions taken by farmers, which need

to be explained in order to assess the damage (Brémond et al., 2013; Brémond, 2011; Durant et al., 2018; Priest et al., 2021a). For this purpose, a process-based modelling approach seems to be the most promising. As experimental data on flood damage on farms are scarce and context-specific (Satrapa et al., 2012), feeding expert knowledge into the models seems most suitable.

In this article, we analyze and discuss the methodological aspects required to develop process-based damage assess-

ment models in a spirit of capitalisation. In particular, we propose a framework for the development of damage assessment models based on expert knowledge. We illustrate the use of this framework around the model **floodam.agri** that we have developed and used to produce flood damage functions for the agricultural sector in France. Two questions are addressed : i/ How useful is the methodological framework we propose for developing flood damage assessment models in the spirit of capitalisation? ii/ What methodological efforts are needed to develop process-based

models that are not only context specific in this capitalisation and cooperation perspective? In section 2, based on a state of the art, we propose a methodological framework for the developpement of process-based models relying on expert knowledge which consist of the four axis i.e : i/ explicit assumptions, ii/ validation, iii/updatability and iv/ transferability. In section 3, the case study, i.e the context and main steps of developpement of **floodam.agri** are presented. Then, in section 4, the four axis and conditions proposed in our methodological framework are tested for

**floodam.agri**. In the discussion 5, the usefulness and limitations of the proposed framework are discussed. Finally, the section 6 concludes by outlining the research avenues to be developed for the improvement of process-based models.

## 2 Methodological framework for capitalizing on modeling efforts

### 2.1 Proposition of a methodological framework

Based on a review of the literature as well as on our own modeling experience, we propose the methodological framework presented in the table 1. It is presented in the form of questions that are as many conditions to be respected for the development of process-based models in a capitalisation perspective. These conditions have been grouped into four main axes which are: i/explicit assumptions, ii/validation, iii/updatability, iv/transferability. We detail the conditions of each axis in the sections 2.2 to 2.5.





**Table 1.** Methodological framework for the development of process-based flood damage models

---

**Axis 1 : Explicit assumptions**

EA1    What are the boudaries and components of the system considered ?

EA2    What are the biophysical processes that cause the damage considered?

*Are the biophysical processes that cause the damage taken into account in the model explicitly considered?*

*Are the biophysical processes that cause the damage implicitly considered identified?*

*Are the links between biophysical processes and flood parameters clearly defined?*

EA3    Which are the assumptions on farmers' decisions?

*Are the links between the farmers' decisions and impacts made explicit?*

---

**Axis 2 : Validation**

V1    Is it possible to compare the model results with sinistrality data?

V2    Is it possible to compare the results of the model with other similar models?

V3    Does the model meet stakeholders' expectations?

V4    Has the model been tested on several application cases?

V5    Has the model been presented and discussed with the experts involved for the development?

*Are modeling assumptions about processes and actions validated with the experts involved?*

*Are the monetization values validated with the experts involved?*

*Are the results of the models validated with the experts involved?*

---

**Axis 3 : Updatability**

U1    Are all the data used in the model and their sources made explicit?

U2    Are the vintages of the data used in the model specified?

U3    Are the data used tracked over time?

---

**Axis 4 : Transferability / improvements**

T1    Are the conditions for adaptations, improvements and transfers described?

T2    Has the model been transferred to another context?

---



## 2.2 Axis 1 : Explicit Assumptions : system boundaries, biophysical processes and decisions

Gerl et al. (2016) reviewed 47 flood damage models (process-based or data driven) in order to create a basis for harmonization and benchmarking. One of their main conclusion is that this requires profound insight into the model structures, mechanisms and underlying assumptions. In the following, we highlight which assumptions need to be explicited.

Flood damage are usually classified in four types : direct tangible (e.g. physical damage due to contact with water), indirect tangible (e.g. loss of production and income), direct intangible (e.g. loss of life) and indirect intangible (Jongman et al., 2012; Merz et al., 2010; Priest et al., 2021b). To evaluate flood damage on economic activities, defining the limits of the system considered is crucial to distinguish between direct and indirect damage since the flood affects not just the property directly affected. As an example, on agricultural assets, Brémond and Grelot (2012) identified induced damage at farm scale due to the links between farm plots and buildings. Nortes Martínez et al. (2020) shows the importance of interactions betweens farms and the cooperative at a winery cooperative scale and the consequences on flood damage estimation. So first, clearly defining the limits and the components of the system under consideration is necessary to avoid problems of double counting or forgetting damage. This refers to the condition EA1 in table 1.

Then, process-based models try to reflect physical or biophysical processes that occurs on the considered system and which generate flood impacts. Those processes are numerous, depend on the component of the system considered and may depend on different flood parameters (Kelman and Spence, 2004). Explicit assumptions on which are the processes considered, on which component of the system and which are the flood parameters involved are essential in process-based models (Davis and Skaggs, 1992). Condition EA2 (table 1) is developed in sub-conditions that helps to detail how the biophysical processes due to flood on the considered system taken into account.

Finally, flood damage results of interaction of flood impacts and human behaviour (Middelmann-Fernandes, 2010). At the end, evaluating the damage in monetary terms requires knowing the repair and restoration choices made by the people affected and their costs. In data-driven modelling those choices are implicitly included in the damage data collected. In process-based models, the property damage avoided technique is used (Shabman and Stephenson, 1996). The repair choices and their costs are hypothetical and fed with expert knowledge. As a consequence, explicit assumptions on the decision rules considered are also critical to properly describe a process-based damage model. This refers to the condition EA3 (table 1).

## 2.3 Axis 2 : Validation

Although the research community has put a lot of efforts into improving flood damage models, Molinari et al. (2019a) point the lack of validation and identify three modalities for the validation of flood damage models which are : i/ the comparison with observed data, ii/the comparison with other models, iii/ the use of expert judgement. In the methodological framework (table 1), the condition V1 questions the possibility to compare the outputs with observed



damage data and the condition V2 to compare the models between them. However, for all sectors, and especially for the agricultural one, a lack of data to fully implement the first modality is commonly observed. As for the second one, a lot of work is being done to compare the different existing models (Gerl et al., 2016; Molinari et al., 2020; Malgwi et al., 2021) in order to have a better idea of the uncertainties. However, the difficulties encountered are often related to the lack of explicit assumptions used in the approaches and modeling choices which brings us back to the importance of properly addressing axis 1 of our methodological framework. As for the third modality, we state that two perspectives must be distinguished: i/the adequacy with the stakeholders' expectations (condition V3) which is related to the use of the model in practice (V4); ii/ the validation with the experts involved in the modelling process (V5). As for the second point, few experience and methodology has been found. Let us mention the experience of Dias et al. (2018) who discussed with experts the data collected for the construction of damage functions on buildings. However, the methodology for validating the models with experts remains to be consolidated. Based on our own experience, we detail in the V5 condition, the sub-conditions which seem to us necessary for the validation by the experts involved in the modeling proces in the following steps: i/discussion of the modeling assumptions about processes and recovery actions ii/discussion of the monetization values iii/discussion of the outputs.

## 2.4 Axis 3 : Updatability

Although some research exists on updating flood hazard models, for example by integrating climate change (Hattermann et al., 2016), the update of flood damage models remains little investigated although necessary (Comiskey, 2005). Updatability is defined as the possibility of updating and should be understood as the anticipation in the modeling process of the possibility of updating the calibration data of the model. This notion is different from the update which corresponds to updating the model outputs. It can be achieved through the updatability of the source data or through simplified methods of actualization of the outputs. The update when it is addressed, concerns the values allowing the monetization as for example, in the last version of the multi-coloured handbook (Priest et al., 2021b). In general, the databases used are rarely made explicit and even less so the vintages. It is therefore important to verify whether the types of data and their sources are made explicit (condition U1, table 1), whether the database vintages used are specified (condition U2), whether the databases are tracked over time (condition U3).

## 2.5 Axis 4 : Transferability

Transferring flood damage model is a challenging issue (Molinari et al., 2020; Jongman et al., 2012; Cammerer et al., 2013). As we dealt with updating in the section 2.4, we focus here on transfer in space and improvements of the model. Improving modelling techniques to transfer data driven flood damage models has been largely explored (Wagenaar et al., 2018, 2021). But, the transfer of process-based model is very challenging mainly beacause it requires a great understanding of origin, calibration, assumptions, field of application which brings back again to the central issue of explicit modelling assumptions (section 2.2). Although process-based modeling approaches seem to be the most promising in terms of transferability, the lack of explicit assumptions hinders this and models developed





remains context-specific. Scorzini et al. (2020) offer an example of transferring and improving a process-based damage model developped for agricultural sector in Italy (AGRIDE-c). AGRIDE-c (Molinari et al., 2019b) relies heavily on **floodam.agri** but the assumptions made for this transfer were not explicit enough. This unfortunate example of non-capitalisation contributed to the motivation for writing this article. It higlights the need to anticipate since the design of the model the different levels of adaptations, improvements and tranfers. Condition T1 (table 1)checks whether the adaptation, improvement or transfer conditions have been taken into account and described at the time of the model design. Condition T2 refers to the effective transfer of the model.

## 3 Case study : the development of floodam.agri in France

### 3.1 Context of development and implications

In France, since 2011, it is mandatory for local communities to conduct cost-benefit analysis (CBA) of their flood management projects, to make them eligible for financial support from the State. Meanwhile, as a support, the French Ministry in charge of Environment proposed a methodology to fulfil CBA (Rouchon et al., 2018) and a working group including researchers and engineers developed flood damage functions. They are available online[1]. Since 2013, over 200 flood management projects have been analyzed by cost-benefit using this method and flood damage functions.

Like for many other countries, this methodology is based on the estimation of flood damage. However, existing models to estimate flood damage were judged not convenient for a national-wide use. As a consequence, the French Ministry in charge of Environment launched studies to develop damage models for different sectors, such as: residential sector, public infrastructures, agricultural sector, and commercial and industrial sector. In this article, we focus on our contribution to produce damage functions for the agricultural sector through the development of the model **floodam.agri**. However, the methodology for all sectors share the same principles: no sufficient data from past events were available to build damage models on a statistical analysis, so process-based modelling approaches have been adopted and they were fed with expert knowledge.

This development context has led to particular requirements. The Ministry needed ready-to-use French National Damage Function but the damage functions should be applicable and explainable to the various stakeholders who would use it at the watershed scale to evaluate their projects. This has resulted in two specific requirements that we have kept to during the development of **floodam.agri**: i/ explicitly explain the assumptions made, ii/ validate the assumptions and outputs at national scale. Since the flood damage functions were intended to be used by local practionners on the long-term use, two specific requirements were added: i/ the use of existing data and open sources if possible updatable, ii/ the possibility to adapt/transfer to specific local contexts.

---

[1] https://www.ecologie.gouv.fr/levaluation-economique-des-projets-gestion-des-risques-naturels





## 3.2 Overview of French National Damage Functions

The database used to locate agricultural assets in France is the Graphical Plot Register (GPR). It lists the agricultural parcels according to a defined typology. In France, there is no database for the census of agricultural buildings. The damage functions produced with **floodam.agri** have been built to be compatible with this. They indicate the estimated expected value of damage in euros by hectare, depending on the water depth, submersion duration, season of occurrence of the flood, and flow speed. Using **floodam.agri**, damage functions were produced for 15 of the 28 sorts of crop of the GPR typology. These 15 sorts accounted for 89% of agricultural areas located in flood-prone areas in metropolitan France in 2010, according to the GPR database. The maximum expected damage is the lowest by hectare for sunflower crops (1 611 Euros) and the highest for arboriculture and orchards (93 549 Euros) (table 2).

For illustrative purpose, the figure 1 shows the damage function of the soft wheat. The damage increases with the flow speed, the submersion duration, and the water depth. It is generally the highest in spring and the lowest in winter.

The threshold effects in the relationship between the damage and the water depth correspond to the water depths at which new types of plant organs are reached by water (e.g. leaves, fruits).

## 3.3 Developement process

The development of **floodam.agri** followed six stages (figure 2).

### The conceptual framework

As described at the top of the figure 4, a crop category is broken down into elementary components. For each component, the damage is estimated based on the biophysical processes at work due to the flood and the actions carried out by farmers after the flood.

### Surveys with agricultural experts

To inform the conceptual framework, in particular the biophysical processes and decisions for each elementary component of a crop category, individual surveys with agricultural experts were carried out. A questionnaire was designed and structured in two parts in order to collect information on the one hand on impacts on farm components and on the other hand, on consequences on farmers' practices. Prior to every interview, production cycles in terms of physiological stages and agricultural work calendar were established based on litterature, for the categories of crop corresponding to the expert interviewed. This information was presented and discussed with the experts too.

This questionnaire was used to conduct semi-structured interviews with 30 experts working in regional technical institutes for agriculture. They were selected according to their area of expertise in terms of families of crops, geographical location. The experts usually had expertise at the level of a crop family that encompasses several categories (table 3). Some had expertise in several families. Among the experts, six were specialists in grain and

**Figure 1.** Example: the flood damage function of the soft wheat



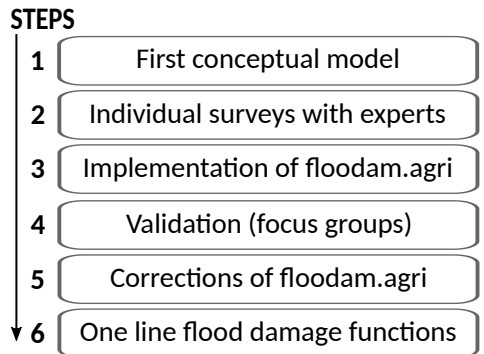

**STEPS**

| | |
|---|---|
| 1 | First conceptual model |
| 2 | Individual surveys with experts |
| 3 | Implementation of floodam.agri |
| 4 | Validation (focus groups) |
| 5 | Corrections of floodam.agri |
| 6 | One line flood damage functions |

**Figure 2.** Development process of the national flood damage functions for agriculture

oleaginous crops, eight in vegetable crops, four in vines, three in fruit-trees, and eight in meadows and feeding crops. The experts worked in geographical areas where crops had been impacted by at least one flood since 2005. We focused on five areas that differ in terms of hydrological and agricultural contexts (see Figure 3): two Mediterranean areas, an area composed of alluvial plains and mountains, an oceanic area, and a rural area composed of plains and plateaus.

**floodam.agri implementation**

Several steps were necessary to produce damage functions with **floodam.agri** (figure 4). The crops for which damage can be estimated with **floodam.agri** are defined in a three-level classification (table 3). The level 1 corresponds to five crop families. It brings together 24 categories of crops usually grouped in agronomy. However, this level is not fine enough to define homogeneous damage processes. The crop category (level 2) is the level where damage process is homogeneous. The crop sub-category (level 3) represents a total of 53 crops that can be related to the second level. For instance, winter wheat, barley, and rye are three types of crops that belong to the winter wheat category and to the grain and oleaginous crops family. All the crops that belong to a same category are associated to a similar vulnerability to floods, but can differ in terms of their other characteristics (yield, selling price, crop calendar, intermediate consumption).

The generic parts of **floodam.agri** are the damaging functions and the actions functions. Damaging functions are the mathematical equations representing the biophysical processes. They associate a proportion of loss or level of deterioration of a component to flood parameters ; for example, for the crop component, damaging functions associate a loss of yield in percent of the standard yield. Action functions models decision rules, which associate behaviours to the proportion of loss or level of deterioration of a component ; for example, for the crop component, actions functions add specific treatments that have to be done to prevent a loss of yield. Action functions are composed of two parts: the farmers' rules to decide whether and how they choose to restore the affected components after a


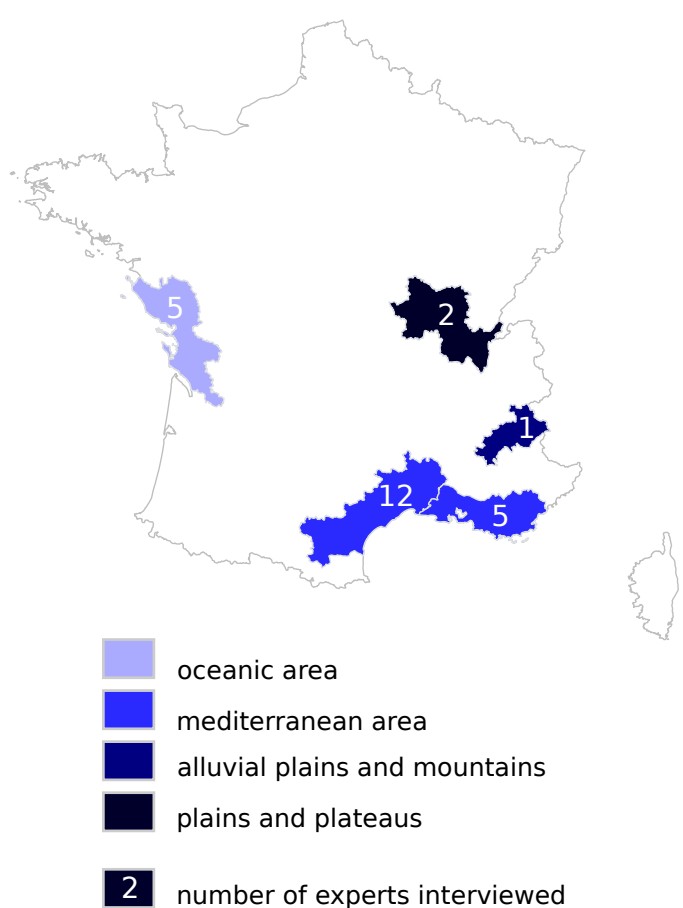

oceanic area

mediterranean area

alluvial plains and mountains

plains and plateaus

2 number of experts interviewed

**Figure 3.** Geographic distribution of the experts interviewed

flood, and the costs of the actions needed to restore the damaged components in terms of expenses and variation in
income.

The mechanisms that lead to the damage to each component are synthesized in figure 5 and detailed in section 4.1.
**floodam.agri** model was implemented using R language.

The flood damage mechanisms modelled in **floodam.agri** are generic and the model needs to be calibrated with
specific local data such as agricultural calendars, yields, and selling prices to produce flood damage estimates for
275 specific contexts. The combination of damaging functions and action functions calibrated enables the production of
damage functions which summs the monetary damage generated by each component. The first damage functions
generated with **floodam.agri** are at the subcategory level (level 3).




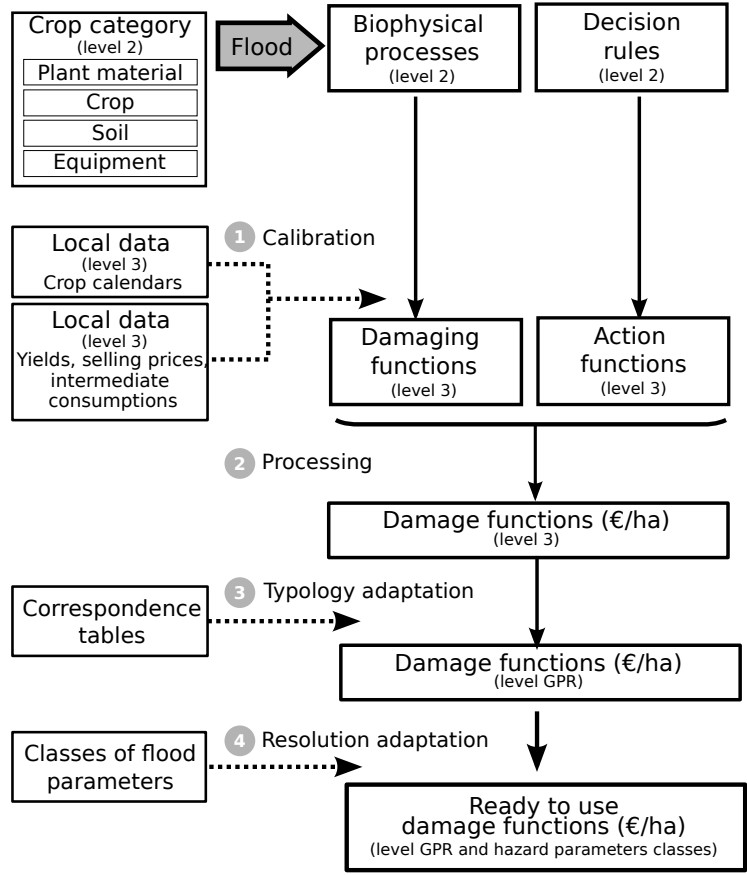

**Figure 4.** Production process of the French flood damage functions with **floodam.agri**

**Validation**

The damage functions at the level 3 are those discussed with experts. The validation process was carried out through focus groups bringing together the experts consulted in individual interviews for each crop family. This steps occured in average one year after the first interview. In total, five focus group have been organised. This step will be detailed in section 4.2.

**Ready to use Flood Damage Functions**

To produce ready to use flood damage functions, two more steps (3 et 4 on figure 4 were achieved : (3) adapting the damage functions to fit the typology used to locate the crops (GPR), (4) adapting the resolution of the functions to fit the available data that pertain to flood parameters.





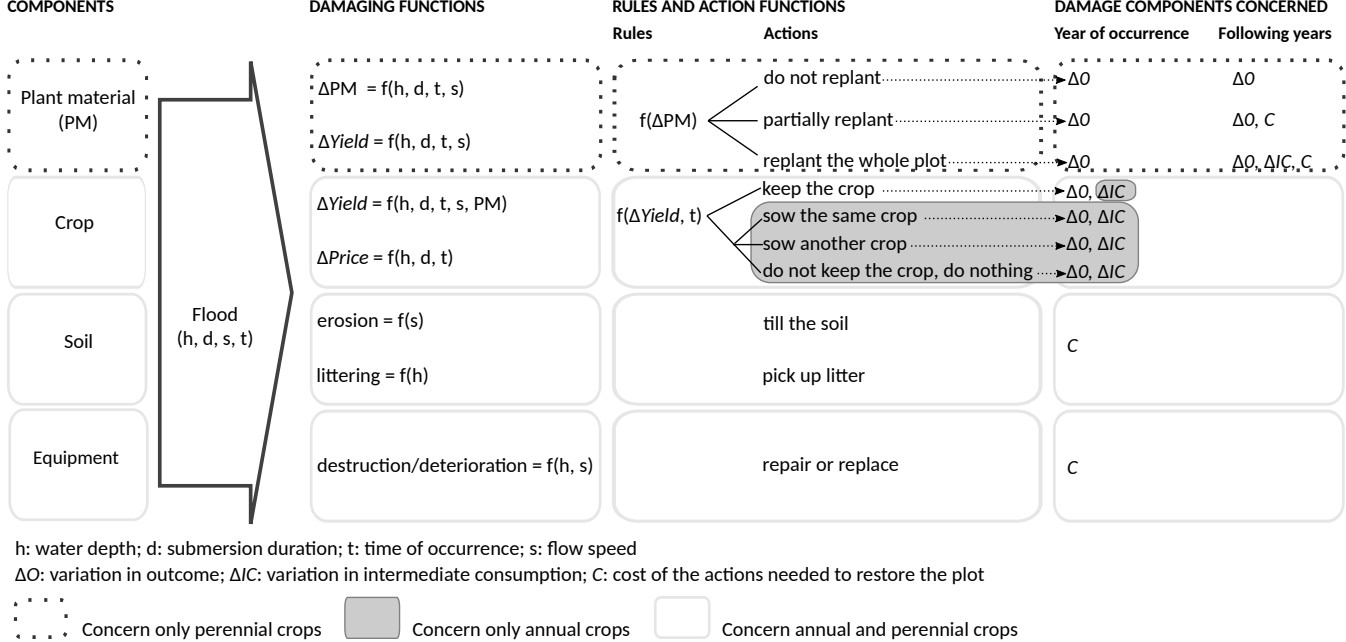

h: water depth; d: submersion duration; t: time of occurrence; s: flow speed
ΔO: variation in outcome; ΔIC: variation in intermediate consumption; C: cost of the actions needed to restore the plot

Concern only perennial crops    Concern only annual crops    Concern annual and perennial crops

**Figure 5.** Description of the generic part of **floodam.agri**

## 4    Application of the methodological framework to floodam.agri

In this section, the methodological framework (table 1) is applied to **floodam.agri**. The objective is to analyze the extent to which the framework makes the modelling process explicit and allows for the transfer of the model to other
study cases.

### 4.1    Explicit assumptions (Axis 1) : the model explicited

**EA1: What are the boudaries and components of the system considered ?**

Flood impacts on the agricultural sector need to be considered through the production process of added value. The figure 6 represents the links between economic entities that may impact the variation of added value. Each economic
entity is composed of physical components (building and parcels) that can be directly affected by a flood and a decision-making entity in charge of production and recovery decisions if a flood occurs. At farm level, the growing process can be impacted either directly by the flood or indirectly if farm's buildings are impacted. In the same way, flood impacts on suppliers may interfers on the production process.

This conceptual framework has been developed based on litterature review and previous work (Brémond et al.,
2013; Brémond, 2011; Nortes Martínez, 2019a). It is important to specify that the transformation can be included in the farm in certain cases. To illustrate this, let's take the example of viticulture. Some winegrowers sell their





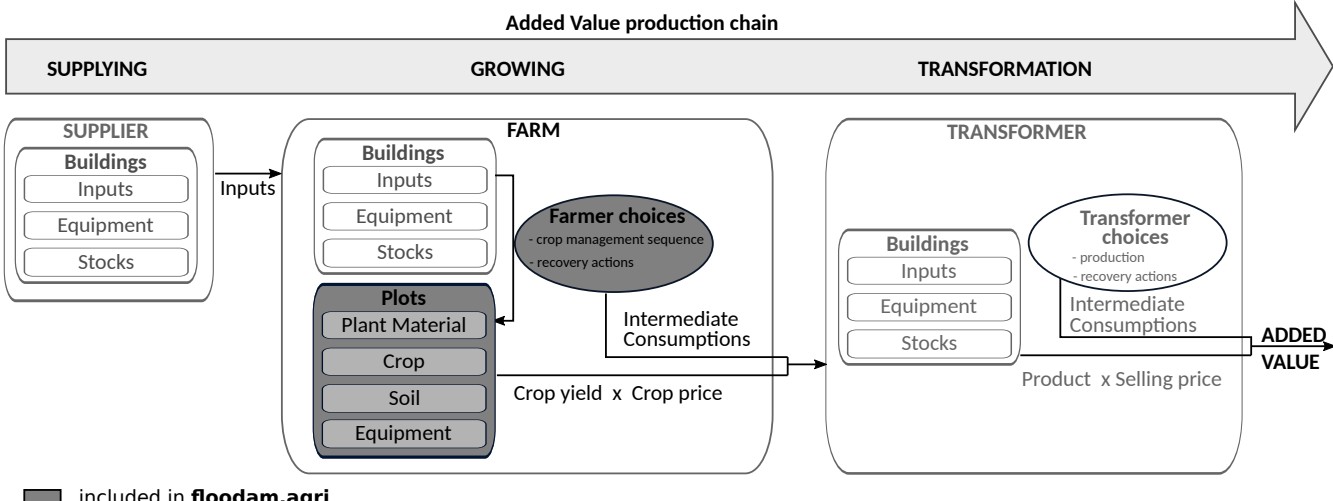

**Figure 6.** Boundaries and components considered in **floodam.agri**

grapes to a cooperative that takes care of the transformation process, while others do the vinification themselves. This considerably modifies the types of impacts to be considered on the systems.

The components in dark grey are those that are considered in **floodam.agri**. It takes into account the physical

components related to the land plots namely crops, plant materiel, soil and equipment on landplots which includes irrigation systems, fences and trellis depending on the crop type. It also takes into account farmer's decision in terms of adaptation of production tasks (crop management sequence) and recovery tasks.

Damage to buildings and their contents (inputs, equipment and stocks) has not yet been taken into account because, for the application of the damage functions, it is currently not possible to locate agricultural buildings in

the existing database. **floodam.agri** also does not consider induced damage at the farm scale i.e damage induced on farm activity due to direct damage on farm equipment for example as evaluated in Brémond and Grelot (2012) or indirect damage at the scale of the area affected by a flood as for example damage propagation on cooperatives as evaluated in Nortes Martínez (2019a) and described in Nortes Martínez (2019b). Indeed, in an operational way, it remains very difficult to obtain information concerning the links between farm buildings and parcels of the same

farm or the links between farms and cooperatives.

Equations 1 to 4 describe the translation of this conceptual framework in economic terms. The total damage to a plot ($D$) is the sum of the costs of the actions needed to restore the plot ($C$) and of the loss of added value ($\Delta AV$). It is calculated as the sum of the damage to each component of the plot ($D_c$): (i) plant material (for perennial crops), (ii) the crop production, (iii) the soil, and (iv) equipment. The crop component is defined as the part of the plant

that is harvested.





The added value is the difference between the outcome of the plot ($O$) and the intermediate consumption due to its management ($IC$). The outcome is the product of the yield ($Y$) and the selling price ($P$), while the intermediate consumption is the consumption in terms of input, material, and labour. The loss of added value is the difference between the usual added value and the added value following a flood.

$$D = \Delta AV + C = \sum_c D_c \tag{1}$$

$$AV = O - IC \tag{2}$$

$$O = Y * P \tag{3}$$

$$IC = Input + Material + Labour \tag{4}$$

$$\Delta AV = AV_{usual} - AV_{flood} \tag{5}$$

**EA2: What are the biophysical processes that cause the damage considered ?**

The methodological framework proposes to discuss this following three sub-questions.

– Are the biophysical processes that cause the damage taken into account in the model explicitly considered?

– Are the biophysical processes that cause the damage implicitly considered identified?

– Are the links between biophysical processes and flood parameters clearly defined?

For each component, the table 4 summarises the processes at work in the formation of damage, the major flood parameters involved, whether the process is taken into account or not in **floodam.agri** and if yes, if the estimation is explicit or implicit and how to estimate the consequences. These processes have been identified based on litterature and during the individual interviews.

The parameters used to characterise the floods are : (i) the height, (ii) the duration of submersion, (iii) the velocity, and (iv) the season. The ranges of values considered for each parameter are indicated in table 5.

We described the time of occurrence of the flood in terms of schedule of physiological stages instead of time of the year to maintain the adaptability of our model to different contexts. It was defined in collaboration with the experts consulted. As an example, the table 6 presents the physiological stages selected at the French scale for winter wheat.

As for the flood height, for each crop, crop height data were also collected (trunk height, fruit height, maximum height).

**Crop**

In **floodam.agri** the main potential impact of floods on the crop component is the loss of yield. But, the table 4 shows that several processes are involved. For example, floods can affect the quality of the crops products which is estimated by a decrease in their selling price.





**Plant material**

The table 4 shows that the main processes that cause plant material i.e tree or vine mortality ar uprooting or asphyxia. The proportion of plants suffering from asphyxia increases with the water depth (because the probability of asphyxia increases with the number of leaves and branches reached by the water) and submersion duration. It also depends on the growth stage at the time of occurrence of the flood (e.g. the roots are less sensitive during dormancy).

Uprooting largely depends on the flow speed. For each process taken into account and for each physiological stage of the crops, the effects are estimated in function of the combination flood parameters.

**Soil**

The flood impacts on the soil component taken into account in **floodam.agri** are erosion and littering (table 4). Erosion depends on the flow speed and the quantity of material carried by flood water depends on the water depth.

**Equipment**

Equipment on the plots (ie irrigation systems, fences, greenhouses, and trellis) can be deteriorated or destroyed (table 4). The deterioration or destruction of equipment depends on the flow speed, that influences the number of devices that move during the flood, and on the water depth that is linked to the number of devices immersed.

**EA3 : Which are the assumptions on farmers' decisions ?**

The assumptions made on the decision rules of farmers after the flood are linked to the damage endured and the physiological stage of the crops. They are explicited for each compoment below.

**Behavior in standard situation**

The behavior of farmers in standard situation is defined by the crop management sequence which is the logical and orderly sequence of tasks that must be performed to achieve the set yield. The periods in which tasks must 370 be performed are defined on the basis of physiological stages. In the example, presented in the table 7, the week numbers indicated correspond to the national adaptation in France.

These sequences of tasks were used as a basis to discuss with the experts the change in farmmers behavior due to flood. The list of potential additional or cancelled tasks is presented in table 8.

**Decisions related to crops**

Faced to a loss of yield of annual crops, farmers decide whether they want to keep the flooded crop. If they decide that it is not worth keeping the crop, they have to choose between three options: they can sow the same crop, sow another crop, or do nothing. Their choice depends on the proportion of the yield that is lost and on the growth stage at the time of occurrence of the flood.





The damage to the crop component relates only to the year of the flood.

Regarding annual crops, in all cases, farmers generally have to modify their usual crop management plan. Thus, the damage depends on the variation in the outcome and expenses.

If farmers decide to keep the flooded crops, the damage is the sum of (i) the loss of outcome due to the loss of yield and the reduction of the selling price, and (ii) the additional expenses in terms of treatments to avoid moisture-related diseases:

$$D_{crop} = \alpha Y_u \times P_u + (1-\alpha) \times Y_u \times \gamma P_u + IC_t \tag{6}$$

with $IC_i$ the additional expenses in terms of treatments, by hectare. If farmers decide to sow the same type of crop, the damage is:

$$D_{crop} = \alpha_2 Y_u \times P_u + IC_s \tag{7}$$

with $\alpha_2$ the yield reduction coefficient that takes into account the fact that late sowing can lead to smaller yields,
and $IC_s$ the intermediate consumption related to sowing, by hectare.

If farmers decide to sow another crop, the damage is the sum of (i) the difference between the outcome of the initial and the new crops, and (ii) the intermediate consumption related to sowing:

$$D_{crop} = Y_u \times P_u - (1-\alpha_2)Y_{new} \times P_{new} + IC_s \tag{8}$$

with $Y_{new}$ the usual yield by hectare of the new crop and $P_{new}$ the usual selling price of the new crop.
If farmers decide to do nothing, the damage is the difference between the loss of outcome and the avoided expenses related to the harvest:

$$D_{crop} = Y_u \times P_u - IC_h \tag{9}$$

For perennial crops, the damage is calculated after taking into account the proportion of plants that are lost (see section 4.1). The damage to crops ($D_{crop}$) is the loss of outcome due to the reduction of the selling price and the
loss of the yield provided by the remaining plants:

$$D_{crop} = (1-\beta)\left[\alpha Y_u \times P_u + (1-\alpha) \times Y_u \times \gamma P_u\right] \tag{10}$$

with $\alpha$ the yield reduction coefficient, and $\gamma$ the selling price reduction coefficient.





**Decisions related to plant material**

In case of loss of plant material, the farmers decide whether or not they want to replant. If they decide to replant,
they then have to choose whether they will replant only the proportion of plants that were uprooted or the whole
plot. These decisions depend on the proportion of plants that are lost. If they replant the whole plot, they have to
uproot the remaining plants after they are harvested. These operations take place during the vegetative rest.

Direct $(D_{PM})$ and delayed $(D_{PM}^d)$ damage to plant material are estimated.

In all cases, the direct damage to plant material by hectare is the loss of outcome due to the loss of plants:

$$410 \quad D_{PM} = \beta \times Y_u \times P_u \tag{11}$$

with $\beta$ the proportion of plants lost by hectare, $Y_u$ the mean usual yield by hectare, and $P_u$ the mean usual selling
price.

If the farmers do not replant, the delayed damage by hectare is the discounted sum of the loss of outcome due to
the loss of plants, for all the years in which the lost plants would have been productive:

$$415 \quad D_{PM}^d = \sum_{i=1}^{A_{max}-A_{PM}} \frac{\beta \times Y_u \times P_u}{(1+r)^i} \tag{12}$$

with $A_{max}$ the usual maximum age of the perennial plants considered, $A_{PM}$ the mean age of the plants at the
time of the flood,[2] and $r$ the discount rate. If the farmers replant only the plants that were lost, the delayed damage
by hectare is the sum of (i) the cost of replanting the proportion of plants lost, weighted by the age of the lost plants,
and (ii) the discounted sum of the loss of outcome until the new plants become productive:

$$420 \quad D_{PM}^d = \beta \times C_{pl} \times \frac{A_{PM}}{A_{max}} + \sum_{i=1}^{A_{prod}} \frac{\beta \times Y_u \times P_u}{(1+r)^i} \tag{13}$$

with $C_{pl}$ the cost of planting one hectare of the perennial plants considered, and $A_{prod}$ the age at which the plants
become productive.

If the farmers replant the whole plot, the delayed damage is the sum of (i) the cost of replanting the whole plot,
weighted by the age of the plants at the time of the flood, and (ii) the discounted sum of the loss of outcome until
the new plants become productive, minus the avoided costs in terms of harvest:

$$D_{PM}^d = C_{pl} \times \frac{A_{PM}}{A_{max}} + \sum_{i=1}^{A_{prod}} \frac{Y_u \times P_u - IC_h}{(1+r)^i} \tag{14}$$

with $IC_h$ the intermediate consumption related to the harvest, by hectare.

---

[2]When calibrating the model, if the mean age of the plants in not known, the assumption that $A_{max} = A_{max}/2$ can be made.




### 4.1.1 Decisions related to the soil and equipement

As for the soil and equipment, the assumption made is that farmers will repair to recover the same state as before
the flood. The repair and replacement actions have been defined with experts in function of flood impacts on the
component. replacement and repair costs

The damage to the soil component ($D_{soil}$) relates only to the year of the flood.

It is equal to the costs of tilling the soil to correct for erosion and picking up litter, which depend on the labour
and mechanisation costs:

$$D_{soil} = (d_{tilling} + d_{cleaning}) \times (C_{labour} + C_{mecha}) \tag{15}$$

with $d_{tilling}$ the amount of time needed to till one hectare of soil, $d_{cleaning}$ the amount of time needed by hectare
to pick up litter, $C_{labour}$ the labour cost, and $C_{mecha}$ the mechanisation cost.

The damage to equipment ($D_{eq}$) relates only to the year of the flood.

It is equal to the replacement and repair costs, which include labour and material costs:

$$D_{eq} = \sum_{i \in I} C_{mat}(i) + \sum_{j \in J} (C_{mat}(j) + d_{repair}(j)C_{labour}) \tag{16}$$

with $I$ the set of devices that need to be replaced, $J$ the set of devices that need to be repaired, $C_{mat}$ the material
cost to replace or repair a device, and $d_{repair}$ the amount of time needed to repair a device.

### 4.2 Validation

In this section, the methodological framework (table 1) is used to describe the validation process implemented for
**floodam.agri**.

### V1: Is it possible to compare the model results with sinistrality data?

As specified in the section 3, up to date, it is not possible to compare flood damage models developed for the
agricultural sector with sinistrality data since no such data exists. In France, sinistrality data on the agricultural
sector are very limited and unsuitable for comparison with the damage functions developed. Indeed, the penetration
rate of private insurance is very low. Compensation is mainly paid through the National Agricultural Risk Guarantee
Fund (FNGRA). However, this system compensates only part of the crop losses (for example, losses of grapes or
cereals are not covered) and, moreover, it is a compensation system based on a declarative estimate of losses at the
time of the flood. It does not take into account, as we have tried to do in this study, the deferred losses and the
variations in expenses linked to farmers' decisions.





**V2: Is it possible to compare the results of the model with other similar models?**

Up to date, no comparison of **floodam.agri** has been done with other models. In France, no other flood damage model for agriculture exists. Comparing **floodam.agri** with existing flood damage model for agriculture such the flood damage functions developed by the FHRC in UK or Agride-C in Italy would required a common case study. No such initiative has been done yet. We are convinced that the effort of explicitness made in this article contributes to go in this direction

**V3: Does the model meet stakeholders' expectations?**

**floodam.agri** was used to produce the damage functions that are recommended for the realisation of the Cost Benefit Analysis (CBA) which are mandatory in France for projects over 2 M euro. Almost 200 CBA have been carried out using flood damage functions produce with **floodam.agri** since 2014. This prouves that **floodam.agri** has met the expectations of the stakeholders involved in the process namely the Ministry of the Environment, the local authorities in charge of the project, the consulting firms that carry out the CBA.

**V4 : Has the model been presented and discussed with the experts involved for the development?**

This condition is specific to process based model approach relying on expert knowledge and from our experience, we proposed three subconditions to be checked :

- are modelling assumptions about processes and actions validated with the experts involved?

- are the monetisation values validated with the experts involved?

- are the results of the models validated with the experts involved?

One of the challenges was to explicitly discuss the assumptions that were developed on the basis of the individual interviews . This required a major effort to illustrate the different assumptions. As for exemple, the figure 7 shows the illustration that was used to present how we modelled the loss of yield of wheat in function of flood paramaters. The following topics were discussed :

- the biophysical processes considered for each component

- the ranges of yield loss in function of flood parameter

- the determination of impacts for each components in function of flood parameter

- the farmers' strategies for crop continuation

- the additional or cancelled tasks and as a consequence the variation in crop expenses

- the replanting strategies




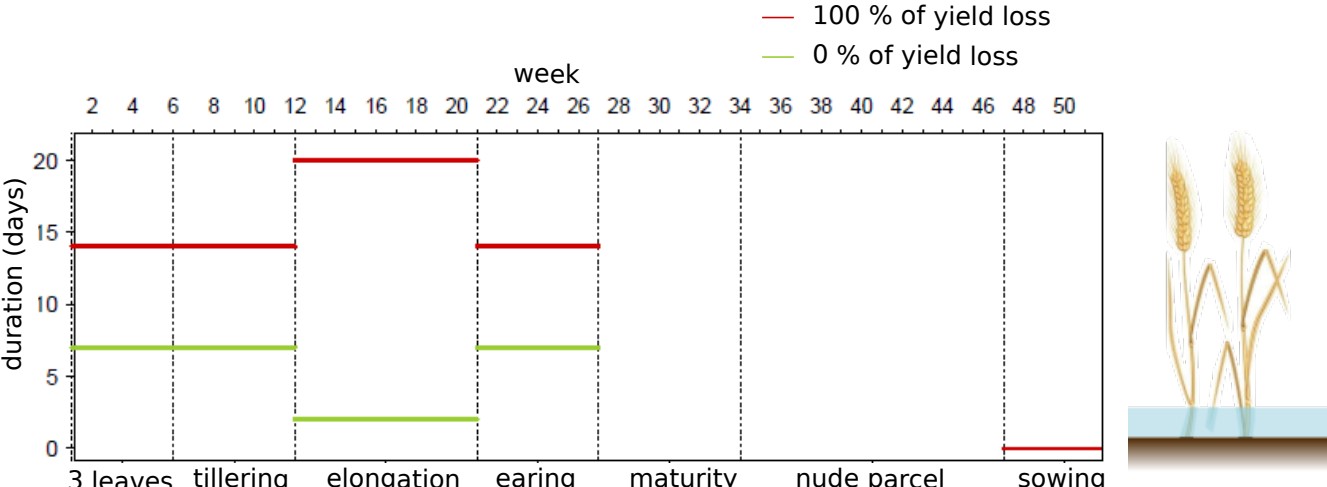

**Figure 7.** example of illustrations used during the validation process

- the list of recovery tasks and their estimated cost (hours of work, equipment)

Each assumptions has been discussed until all experts agreed to validate them. Where the validated assump-
tions were different from those we had presented, we have corrected them. Condition V4 is fully accomplished for
**floodam.agri**.

### 4.3 Updatability : the origin and the vintage of the data specified

In this section, the methodological framework (table 1) is used to describe the updatability of **floodam.agri**.

**U1 : Are all the data used in the model and their sources made explicit?**

To produce Flood Damage Functions, **floodam.agri** requires : i/ an estimate of usual yields,ii/ an estimate of selling
prices, iii/ an estimate of intermediate consumptions, iv/ physiological stages and crop management sequence. The
table 9 lists all the data and their source used in **floodam.agri**. There is no homogeneous database that provides
information on all the technical and economic data of the crops. We had to collect this information from different
databases depending on the crop and sometimes complete this information based on expert opinion. It is therefore
all the more important to be rigorous about making the data used explicit.

**U2 : Are the vintages of the data used in the model specified?**

The vintage used and the frequency of update are specified in the table 10. Since the databases used are heteroge-
neous, the vintages of the databases are also heterogeneous.





**U3 : Are the data used tracked over time?**

The table 10 shows the update frequency of the databases used. Updating the data that is published annually is easy. On the other hand, to update data from documents whose publication frequency is not predetermined requires checking for each data if a new edition has been produced. If not, a validation with experts should be renewed.

To sum up, tables 9 and 10 shows that the updtability of data is not homogeneous. Three modalities can be distinguished :

– input data come from a single database which tracked over time (eg yields)

– input date come from different databases with different update frequencies (eg selling prices and intermediate consumptions)

– input date come from expert knowledge (eg physiological stages)

### 4.4 Transferability

In this section, the methodological framework (table 1) is used to describe the conditions on transferability.

**T1 : Are the conditions for adaptations, improvements and transfers described ?**

The possibility to adapt **floodam.agri** to different contexts was a requirement. Then, it has been anticpated in the modelling process. The different steps for adaptation from the simplest to the most demanding are identified according to the differences between the context in which **floodam.agri** was developed and the context in which it could be transferred. Methodological proposals are made for each of these levels. The levels of adaptation are showed in the development process of **floodam.agri**(figure 8).

**Adjusting damage functions resolution**

The first level of adaptation (figure 8) concerns the compatibility between the flood damage functions produced with **floodam.agri** and existing hydraulic and hydrological models in terms of with resolutions (step 1 in 8).

Damaging functions were built with a resolution of one week in terms of time of occurrence, and one day in terms of submersion duration (see Table 5). In practice, because these parameters are often available with a lower resolution, we adapted the damage functions accordingly. Four categories of submersion duration have been defined (low, medium, high, and very high) with the correspondence given in table 11. Four categories of time of occurrence, which correspond to the four seasons, have also been defined with the correspondence indicated in table 12. To adapt the damage functions to the new categories of parameters, we averaged the values of damage that belong to a same category. This implies that we assumed a uniform distribution of the damage within each category.

**floodam.agri** can generate flood damage functions with a higher resolution easily.



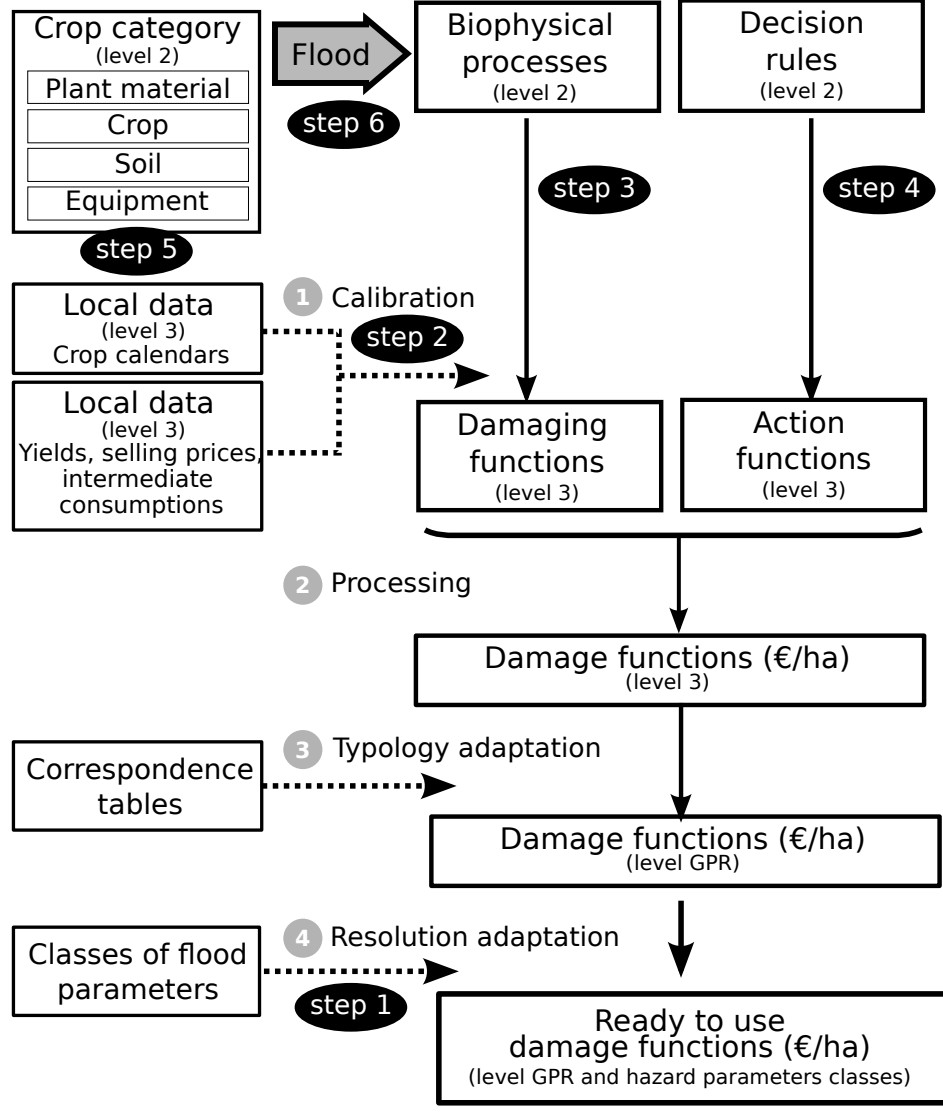

**Figure 8.** Steps of adaptation to transfer **floodam.agri**

**Adjusting to local data**

The second step concerns the adjustement of local data (local yields or selling prices). It is necessary to ensure that
data listed in section 4.3 exists on the study area. The problems encountered in this case may be related to the
typology of crops that will have to be adapted too.





**Adjusting to the climate variation**

If only the climate is different, the timing of the physiological stages (table 6) of each crop will have to be adapted (step 3 on figure 8). Since the physiological stages have been calibrated on a weekly basis, these calendars can be
adapted to a new context on the basis of existing bibliographical and technical data on the area of application and/or on the basis of interviews with agricultural experts, taking care to cover the diversity of crops. The calibration of the physiological stages will lead to the updating of the damaging functions.

### 4.5   Adjusting cultural practices and behaviours in case of flooding

If in the context of application, particular cultural practices or behaviors of farmers in case of flooding exist, an
adaptation of the action functions will be necessary (step 4 on figure 8). This will be done by updating on the one hand the crop management sequence (actions planned in normal situation) and on the other hand, farmers decision rules in case of flooding. This adaptation require collecting agronomic data and/or expert knowledge.

**Adding a new crop**

If a crop is to be added to the list of 53 existing crops in **floodam.agri**, two options should be considered. First, it is
necessary to determine whether the crop can be assigned to a vulnerability category. If so, it is necessary to calibrate the damaging and action functions with the physiological stages, crop management sequence, yield and price of the crop. If not, it will be necessary to create a new crop category (step 5 on figure 8) and to develop new damaging and action function. For this, data collection from agricultural experts will be necessary. Moreover, agroeconomic data will have to be collected to calibrate the functions.

**Taking into account new hazard parameters**

This is the most important level of adaptation because it requires to repeat for each crop category all the biophysical processes and the impact on farmers' behaviors in order to produce new damaging and action functions (step 6 on figure 8). This type of transfer necessarily requires work with experts. However, once the new damaging and action functions are produced, it is possible to apply the rest of the process with the same agro-economic data.

**T2 : Has the model been transferred to another context ?**

To date, some adjustments have been done to adjust resolutions (step 1) or to adjust local data (step 2) in the frame of the mandatory CBA of flood management projects. In Mao (2019), an adaptation a flood damage functions has been done at regional level (step 2) using regional data. Work is underway to adapt **floodam.agri** to coastal flooding (step 6).





## 5 Discussions

### 5.1 A crucial contribution to the clarification of assumptions

The proposed framework clarifies the components, interactions and decision entities that are or are not considered in the damage assessment model. In economic systems, added value is produced on spatial entities (plots in the agricultural case) and depends on production factors (material, labor, input) and decision rules. In the case of agriculture, the added value increases on the plots and is then stored and transformed in other spatial entities on or off the farm. Nortes Martínez et al. (2020) shows the importance of these interactions for avoiding over or understimate in damage assessment. Because of the complexity of these mechanisms of localisation of added value in a production chain, the FHRC recommends, in an operational way, not to take into account the indirect effects (Penning-Roswell et al., 2005). However, making the limits of the modeled system explicit remains fundamental in the classification of damage between direct and indirect. The larger the system considered, the more it will include effects that could be considered indirect. Developing models that locate and characterise interactions between several components in the field is time demanding. Depending on operational needs, this approach may be required (resilience analysis of a sector affected by a project) or not (large-scale damage assessment on all the issues).

From the modeling experience presented in this article around **floodam.agri**, the proposed framework concerning the explicitation of assumptions appears to us to be effective for two main reasons. Firstly, the explanation of the assumptions facilitated the collection of information from the experts. Indeed, we found that the logic we proposed to deconstruct the biophysical processes and the decisions made by farmers was consistent with the cognitive approach of damage assessment of the experts. In this sense, the application of the framework reduces the uncertainties surrounding the collection of expert knowledge. Secondly, the explicitness of the assumptions appears to be a necessary condition for the implementation of the other axes, namely validation, updatability and transferability.

This effort to clarify assumptions is also necessary for continuous improvement. In this sense, although the inclusion of farmers' decisions in damage modeling has been improved significantly in **floodam.agri**, we suggest ways to continue in this direction. The farmers' behavior represented in **floodam.agri** is a standard behavior. Collecting data from agricultural experts who have witnessed flooding on a large number of farms allows us to model this standard behavior. However, we must be aware and vigilant that this average view does not reflect the diversity of individual vulnerability situations at the farm level. Thus, at the individual scale, decisions, especially those concerning long term issues such as replanting, will depend on individual parameters such as investment dynamics, the age of the farm manager, the farm's trajectory... Furthermore, **floodam.agri** does not take into account adaptation decisions that could be made at the time of reclamation, such as changing the crop. Understanding the internal and external determinants of adaptation implementation would require a different approach and investigation at the individual level.





## 5.2 Consolidate the validation

The proposed framework allows for a clear improvement in the validation methodology with experts. However, we are aware of the need to consolidate this aspect. Two avenues could be considered: On the one hand, the comparison of model results with each other and on the other hand, the comparison with claims data (Molinari et al., 2019a). We consider that the clarification of the assumptions is a prerequisite for both avenues and the framework presented here is a step towards the possibility of comparing models with each other. Concerning the collection of ex post damage data, in particular for the agricultural sector, this is a real challenge that requires a long-term effort. Some interresting intiatives are to be higlighted, as for example, the validation carried out by Chau et al. (2015) or Shrestha et al. (2021). The methodology is key and requires the realisation of feedback with a reproducible data collection format. The implementation of observatories appears to be a major priority.

## 5.3 Capitalise over time with updatability

The proposed methodological framework requires the specification of all the data used, their source and their vintage. This makes it possible to consider updating the models produced for a given context over time. This is the case, for the damage functions produced thanks to **floodam.agri**. On the other hand, this effort makes it possible to consider the transfer by comparison of existing databases from one context to another. A difficulty persists for data that are not tracked over time, and in this case we recommend either updating the data on the basis of expert opinions, or using a discount rate whose value must be specified.

## 5.4 Anticipating the transferability to capitalise in space

Although **floodam.agri** has not yet been transferred to other cases of study than France, we highlight that this property has been anticipated. The proposed methodological framework allows us, right from the design stage, to be in line with this spirit of capitlisation and addition of modular bricks.

## 6 Conclusions

Process-based flood damage assessment models relying on and expert knowledge are widely researched and used operationally. However, it is often observed that this work cannot be capitalised on because the models are too attached to their development context. In this paper, we state that process-based models are not doomed to be context specific if the process of data collection and explanation of modeling assumptions is rigorous. We propose a framework that improve the developpement of process-based flood damage models by meeting the properties of assumptions explicitness, validation, updatability and transferability. We show that respecting these properties could help structure a common modeling effort at the international level.



By applying the proposed methodological framework to **floodam.agri**, we show that it is possible to describe explicitly the modeling assumptions. Given the complexity of the phenomena (biophysical and decisional processes), the diversity of the data sources, we argue that the methodological framewrok is useful to structure and anticipate since the begining of the development process a spirit of capitalisation in time and space. This rigorous work is a necessary condition to consider the possibility of improvement in the long term and of cooperation around the development on an international scale. The framework proposed here thus opens up prospects for cooperation in improving and transferring existing models, particularly agricultural ones. In terms of research, this work of methodological improvement must be carried out in parallel with the improvement of data collection on the impacts of floods in terms of monetary damage but also to improve the understanding of biophysical damage processes and repair decisions.

*Code and data availability.* **floodam.agri** has been implemented in R language and will soon be available as a package.

*Author contributions.* PB, ALA and FG developed the conceptual model and collected expert knowledge. FG and ALA implemented it in R language. PB wrote a first complete version which was reviewed by all authors.

*Competing interests.* The authors declare having no competing interests.

*Acknowledgements.* This work benefited from the support of the French Ministry of Environment and KIM WATERS MUSE (project moom-agri).





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





**Table 2.** Categories of crops in the RPG database, area in flood-prone areas, and maximum damage estimated with **floodam.agri**

| Category | Area in flood-prone areas (ha) | Maximum damage (Euros/ha) |
|---|---|---|
| No information | 1 572 | - |
| **Soft wheat** | 5 336 421 | 2 109 |
| **Grain and silage corn** | 3 067 195 | 1 897 |
| **Barley** | 1 595 271 | 1 927 |
| **Other cereals** | 1 119 601 | 1 658 |
| **Rapeseed** | 1 525 055 | 2 154 |
| **Sunflower** | 713 633 | 1 611 |
| **Other oleaginous** | 76 743 | 1 736 |
| Protein crops | 372 320 | - |
| Fibre plants | 47 354 | - |
| Seeds | 72 248 | - |
| Set-aside lands (without production) | 0 | - |
| Industrial set-aside lands | 0 | - |
| Other set-aside lands | 402 587 | - |
| Rice | 25 721 | - |
| Grain legumes | 14 770 | - |
| **Fodder** | 176 884 | 2 544 |
| Pasture | 1 888 703 | - |
| **Permanent grasslands** | 6 488 945 | 2 067 |
| **Meadows** | 3 665 000 | 2 135 |
| **Orchards** | 87 890 | 93 549 |
| **Vineyards** | 449 947 | 50 887 |
| Shell fruits | 26 117 | - |
| Olive trees | 10 990 | - |
| **Other industrial crops** | 431 726 | 2 152 |
| **Vegetables - Flowers** | 331 381 | 20 783 |
| Sugar cane | 0 | - |
| **Arboriculture** | 4 204 | 93 549 |
| Miscellaneous | 298 808 | - |
| **TOTAL** | 28 231 555 | 93 549 |

The areas in flood-prone areas were estimated using the approximate potential flood extent (EAIP), which was estimated for the whole country within the frame of the first national flood risk assessment between 2011 and 2017. The maximum values of damage are calculated taking into account all possible combinations of flood parameters. The categories in bold are linked to a damage function produced with **floodam.agri**



**Table 3.** Families and categories of crop available in **floodam.agri**

| Family (level 1) | Category (level 2) | Sub-category (level 3) |
|---|---|---|
| Meadows and feeding crops | Meadow | Meadow |
| | Recently sowed meadow | Recently sowed meadow |
| | Alfalfa | Alfalfa |
| | Recently sowed alfalfa | Recently sowed alfalfa |
| Grain and oleaginous crops | Corn | Corn |
| | | Non food corn |
| | | Sorghum |
| | | Grain corn |
| | Silage corn | Silage corn |
| | Winter wheat | Winter wheat |
| | | Barley |
| | | Non food wheat |
| | | Silage wheat |
| | | Triticale |
| | | Durum wheat |
| | Spring wheat | Spring wheat |
| | | Spring barley |
| | | Spring durum wheat |
| | | Spring oat |
| | | Grain spring wheat |
| | Rape | Rape |
| | | Non food rape |
| | | Oleaginous |
| | Sunflower | Sunflower |
| | | Non food sunflower |
| | | Silage sunflower |
| Fruit trees | Apple tree | Apple tree |
| | Pear tree | Pear tree |
| | Cherry tree | Cherry tree |
| | Peach tree | Peach tree |
| | Apricot tree | Apricot tree |
| | Plum tree | Plum tree |
| Grape vines | Wine grape | Wine grape |
| Vegetable crops | Asparagus | Asparagus |
| | Salad | Salad |
| | Field tomato | Field tomato |
| | Greenhouse tomato | Greenhouse tomato |
| | Various field vegetables | Melon |
| | | Carrot |
| | | Onion |
| | Tied-in vegetables | Eggplant |
| | | Pepper |
| | Greenhouse tied-in vegetables | Cucumber |





**Table 4.** Biophysical processes considered or not in **floodam.agri**

| Biophysical processes | Flood parameter | taken into account | Estimation |
|---|---|---|---|
| **Crops** | | | |
| poor flowering or fruiting by root apshyxia | season, duration, height | explicit | loss of yield |
| destruction of buds, flowers, fruits by contact | season, duration, height | explicit | loss of yield |
| increase in cryptogamic diseases | season, duration, height | explicit | loss of yield |
| growth alteration by root apshyxia | season, duration, height | explicit | loss of yield |
| growth alteration by crop laying down | velocity, height | explicit | loss of yield |
| growth alteration by leaf asphyxation | season, sediment, height | explicit | loss of yield |
| growth alteration by salinity | season, salinity | no | |
| growth alteration by contamination | season, contimation | no | |
| excess of water in the fruits | season, duration, height | explicit | price decrease |
| soiled fruits by sediment deposit | season, sediment, height | explicit | loss of yield |
| soiled fruits by contamination | contamination | no | loss of quality/yield |
| **Plant material** | | | |
| mortality by uprooting | velocity, height | explicitly | replantation strategy |
| mortality by root asphyxia | season, duration, height | explicitly | replantation strategy |
| mortality by leaf asphyxia | sediment, height, duration | implicitly | replantation strategy |
| mortality by salinity | salinity | no | |
| mortality by contamination | contamination | no | |
| **Soil** | | | |
| deposits of debris and waste | velocity, height | explicit | repair costs |
| erosion without loss of soil | velocity, height | explicit | repair costs |
| erosion with loss of soil | velocity, height | explicit | repair costs |
| soil contamination | contamination | no | |
| soil salination | salinity | no | |
| **Equipment** | | | |
| pulling out and moving irrigation pipes | height, velocity, season | implicitly | pipe reinstatement |
| fence degradation and debris build-up | height, velocity | explicitly | cleaning and repair costs |
| trellising torn off by the current | height, velocity | implicitly | replacement |
| damaged trellising | height, velocity | explicitly | repair costs |



**Table 5.** Ranges and resolution of the flood parameters used in **floodam.agri**

| Parameter | Categories | Range | Resolution | Unit |
|---|---|---|---|---|
| water height | - | 0 to 250 | 10 | cm |
| submersion duration | - | 0 to 20 | 1 | day |
| velocity | low, medium, high, very high | 0 to 0.5; 0.5 to 1; 1 to 2; > 2 | - | m/s |
| season | crop growth stages | - | - | |

**Table 6.** Distribution of the physiological stages of wheat over the weeks of a year

| physiological stage | week |
|---|---|
| sowing | 47 |
| emergence | 52 |
| three leaves | 2 |
| tillering | 6 |
| stem elongation | 12 |
| earing | 20 |
| maturity | 27 |
| nude parcel | 34 |

**Table 7.** Distribution of the crop management sequence on the weeks of the year for wheat

| task | 40-43 | 44-48 | 49-52 | 1-4 | 5-8 | 9-13 | 14-17 | 18-21 | 22-26 | 27-30 | 31-34 | 35-39 |
|---|---|---|---|---|---|---|---|---|---|---|---|---|
| soil ploughing | x | | | | | | | | | | | |
| sowing | | x | | | | | | | | | | |
| fertilising | | | | | x | | x | | | | | |
| treatment / weeding | | x | | x | x | | | | | | | x |
| harvest | | | | | | | | | | | x | |





**Table 8.** List of additional or cancelled tasks taken into account in **floodam.agri**

| additional | cancelled |
|---|---|
| sowing | |
| oversowing | |
| treatment | treatment |
| chemical harvest | harvest |
| replanting | |

**Table 9.** Data sources

| Type of estimates | Sources for: | | | | |
|---|---|---|---|---|---|
| | Meadows and feeding crops | Grain and oleaginous crops | Fruit trees | Grape vines | Vegetable crops |
| Localisation | GPR | GPR | GPR | GPR | GPR |
| Yields | AAS | AAS | AAS | AAS | AAS |
| Prices | SADs | ASB | IPPAC | LR data | IPPAC, SADs |
| Harvest | experts | SADs | SADs, LR data | experts | SADs |
| Sowing/Plantation | experts | experts | SADs | SADs | expert |
| Treatments | - | - | Eco-Phyto 2018 | Eco-Phyto 2018 | experts |
| Crop calendars | LR data, experts | LR data, experts | LR data, experts | LR data, experts | LR data, experts |

GPR: Graphical Plot Register; AAS: Annual Agricultural Statistics database; SAD: Scales of Agricultural Disasters; ASB: Agricultural Situation Bulletin; IPPAC: Index of Producer Prices of Agricultural Commodities; LR data: technical and economic memento of the main agricultural productions in Languedoc-Roussillon and fact sheets on the Languedoc-Roussillon region

**Table 10.** Vintage and update frequency of database used to apply **floodam.agri** at the national scale in France

| data | database | vintage used | update frequency |
|---|---|---|---|
| localisation | GPR | 2010 | annual |
| yields | AAS | 2009, 2010, 2011 | annual |
| price | IPPAC | 2009, 2010, 2011 | annual |
| price | ASB | 2009, 2010, 2011 | annual |
| price | SADs | 2007 | occasional |
| price | TEMMAPL | 2012 / experts | occasional |
| IC | SADs | 2006, 2007 / experts | occasional |
| IC | TEMMAPL | 2012 / experts | occasional |
| IC | Eco Phyto | 2018 | occasional |
| physiological stages | TEMMAPL | experts | occasional |
| crop management sequence | TEMMAPL | experts | occasional |




**Table 11.** Categories of flood duration for the French flood damage functions

| Category | Minimum (Number of days) | Maximum (Number of days) |
|---|---|---|
| low | 0 | 1 |
| medium | 2 | 4 |
| high | 5 | 10 |
| very high | 11 | 20 |

**Table 12.** Categories of time of occurrence of the flood for the French flood damage functions

| Category | Beginning (week of the year) | End (week of the year) |
|---|---|---|
| Spring | 14 | 26 |
| Summer | 27 | 39 |
| Fall | 40 | 52 |
| Winter | 1 | 13 |