# Peer review of "Process-based flood damage modelling relying on expert knowledge: a methodological contribution applied to agricultural sector"

_Natural Hazards and Earth System Sciences, 2021_

## Author Response (AR1)

**Process-based flood damage modelling relying on expert knowledge: a methodological contribution applied to agricultural sector**

Point by point answer of the authors to referees comments

Pauline Brémond et al

2022-06-29

**1 Summary of changes regarding observations and recommendations from referee 1**

We thank referee 1 for his review of our paper. We particularly appreciate the critical and detailed analysis he made. We think that taking these remarks into account has brought real added value to the paper. In particular, this allowed us to provide more illustrations of the modelling steps and better highlight the work we have done. In the following, we have listed all the remarks, questions and suggestions made by referee 1 and numbered them from 1 to 10 for general comments. Sometimes comments have been subdivided into sub-sections in order to respond specifically to each suggestion. This is the case for general comments 7 and 9. Referee 1's comment is given in bold and how we took this into account in the new version of the article is given in plain text.

**General comment 1 : The manuscript deals with the modelling of flood damage to agriculture, one of the (relatively) less investigated sector among the exposed assets. Although this makes the paper a potentially interesting contribution for NHESS, it actually suffers from several criticalities that, in my opinion, prevent its publication in its current version.**

- Thanks to the comments and suggestions of the two refrerees, a great number of changes have been made to the article that have improved the scope of the work that has been done.

**General comment 2 : I would suggest the Authors to reconsider the way they presented their work, by better emphasizing the modelling aspects related to the proposed tool (floodam.agri), rather than presenting a 'philosophical' paper, with limited usefulness.**

- The statement by referee 1 that this was a philosophical article strongly challenged us because we wanted, on the contrary, to share a reflection on the methodology of developing process-based models that was based on strong field experience and articulated with strong operational needs. As it seemed to us that referee 1 had not perceived the operational scope in our description of the methodological approach and its application, we resumed writing the case study section to clarify this point.

**General Comment 3 : the actual structure of the manuscript makes it very confusing, with repetitions**

- A global re-reading of the article has been done to remove repetitions.

**General Comment 4 : the article does not give the right emphasis on the model itself and on its innovation, which should be the main core of the paper. I would therefore re-arrange the structure of the manuscript by first presenting the model (with more details on it) and then explaining why it should be considered a "good" model and what are its current limitations.**

- The description of the methodology (supplements: questionnaire grid for experts, focus group recommendations), the modelling (illustrations based on the case of apple crop) and the results (supplement

on national damage functions) have been greatly improved in this new version. It seems to us that these additions allow a better understanding of how and for what purpose **floodam.agri** was developed and give a better emphasis to the model.

**General comment 5 : On the contrary, the Authors just presented a description of the "archetype of a model" and then they try to explain (with limited success) why their proposed model should be considered as "the" model.**

- In order to avoid presenting **floodam.agri** as "the right model", we have taken over the article to avoid this being implied. We have highlighted other very effective models that have been developed before, in particular the models developed by the FHRC and the AGDAM model developed by the USACE. We have gone back to the text especially in the case study section to share our experience of the difficulties of reusing process-based models because they were too context specific. We have emphasized in the article that a rigorous methodological framework is needed so that modeling efforts that are being made internationally can be capitalized upon.

**General comment 6: In particular, the first part of the manuscript is very general, since the definition of the methodological framework for the development of a process-based flood damage model (Table 1) is not "new", given that the listed questions are typically taken into account in the development phase of any process-based model.**

- Contrary to what referee 1 indicates, we believe that the development of process-based models rarely anticipates the possibility of transferring or updating the developed model over time. We checked the literature and did not find any methodological articles addressing these aspects. We indicated more explicitly in the case study section that we had chosen to develop a model because we had not been able to transfer existing models (FHRC or AGDAM). We have added a comparison table of these three models (appendix C).

**General comment 7 : More importantly, the way the Authors presented their model is also not very convincing. Indeed, unfortunately, when presenting floodam.agri, they failed in properly addressing some of the questions raised by themselves. I especially refer to the ones labelled as "Axis 1: explicit assumptions", i.e. model and data transparency, which (I agree with them) should be regarded as one of the most important features of a model. I just mention hereafter some examples related to the lack of details on modelling assumptions within the model.**

- We have significantly modified the section "explicit assumptions" to take into account referee 1's comment. We have chosen an example of a crop (apple) to highlight the level of detail on the modeling assumptions and the way experts were involved in the construction of these assumptions.

- **GC 7.1: biophysical process taken into account to develop crop susceptibility functions: if you heavily claim for explicitness, the only Table 4 cannot be considered enough, but I would expect a thorough description on how the crop susceptibility functions were developed from a methodological point of view (e.g. by also considering a few crops as example and explaining how the different processes impact on the produced functions: i.e. how the plots reported in Figure 1 have been derived? (why I observe abrupt changes in damage at certain water depths and duration thresholds? Which are the driving mechanisms for these specific patterns?). I think this would be very interesting to know from a modeler's perspective and this could be a real added value to the manuscript.**

    - The tables 4 and 5 have been added to show how the damage coefficients for plant material and crops were defined with the experts. The links between the processes mentioned by the experts and the defined damage levels are explained for each physiological stage.

- **GC 7.2: interviews with experts: no detailed information is provided within the manuscript on how the Authors actually took advantage of experts' knowledge for developing floodam.agri. Again, I would suggest you to better describe the following aspects: how the questionnaires were structured (templates may also be included as supplementary material), how the collected data were analyzed and, in particular,**

whether you found uncertainty in the collected data and you handled this uncertainty in developing the model. Moreover, (if I understood correctly) the same experts were involved both in the development as well as in the "validation" (I would suggest not using this word) stage of the model: doesn't this introduce a bias in the results of model "validation"?

- This comment has been taken into account on four levels:
  * First, the interview guides used to conduct the individual interviews for the tree experts were translated and added as supplementary material.
  * Second, the tables 4 and 5 have been added to show more explicitly the assumptions that were discussed and agreed upon with the experts.
  * Third, section V4 on the validation of the model has been reworked and in particular the illustration in figure 7.
  * Fourth, the conclusions of the focus group that was carried out for arboriculture and the improvements that were proposed and taken into account for the production of the national damage functions were listed in an supplementary material (3.focus group).

- **GC 7.3: farmers' decisions: I would find interesting to know if you could report more (quantitative) details on the driving factors for farmers' behaviors.**

  - We have reworked the description of the decision strategies taken into account in **floodam.agri**. In particular, the tables 7 and 8 have been added to illustrate more explicitly the links between the damage, the crops concerned and the possible decisions.

- **GC 7.4: the description of damage to soil should be better explained: it is not clear how tilling and cleaning are assessed, as I would expect these to be highly influenced (minimum) by the hazard parameters, soil and crop type. The modelling of damage to the equipment needs to be better clarified as well.**

  - As suggested by referee 1, we added more detail on soil damage using the apple crop case. The table 9 has been added for this purpose.

- **GC 7.5: could you provide details on how the flood impact to the quality of the crops is modelled in floodam.agri?**

  - A detailed and illustrated explanation for the apple case of how quality impact is taken into account is given in section 3 (EA2: What are the biophysical processes that cause the damage considered? , Crop).

**General comment 8 : The Authors may reply to my comments raised above that it would be impossible to provide full details of a (complex) model within a Journal paper, with limited available space. This is true, but, at the same time, you cannot claim a full lack of transparency (as you are experiencing also with your paper) of other models in the literature, by also mentioning this as the reason from writing the present article. So, the Authors' comment on this point (P8-L186-189) appears to me a bit subjective (and not fair), given that the cited Agride-c is a well documented, explicit model, which seems to be originated by the same need for model transparency claimed by the Authors.**

- To better show in the article, all the types of crops treated, we have placed in additional material all the national damage functions produced with **floodam.agri** and recommended in France.
- Concerning Agride-c and its presentation in Molinari et al. (2019) and Scorzini et al. (2020), we did not write that there was a full lack of transparency in presenting this model, this assertion seems to us excessive. It turns out that this model can be seen as a transfer from **floodam.agri** to the Italian context as it is largely based on the work presented in Agenais et al (2013). Indeed, the crop damage coefficients that largely determine the damage are directly adapted from Agenais et al, 2013. Similarly, the crop pursuit strategies and soil damage are very similar to what was proposed in Agenais et al, 2013. We have expressed the idea, without elaborating on it, that this transfer was not carried out in a way that allows for capitalisation. For example, the "validation" part by the local experts consulted is

not documented in a sufficient way to understand what are the limits of the assumptions retained in **floodam.agri** for the Italian context. One possibility is that the description of **floodam.agri** was not explicit enough to discuss these assumptions, which is one of the reasons for writing our article. We made this point more explicit in the new version.

**General comment 9 : Another aspect that would deserve more attention is the actual applicability of the model. The Authors mentioned that floodam.agri has been already applied in France in several projects. It would be then very valuable if you could give more details on these experiences, e.g. which were the main difficulties for application, necessary adaptations and/or assumptions in the input data (for both hydraulic and vulnerability / exposure data). Indeed, floodam.agri, as any micro-scale process-based model, requires very detailed input data, with some of them usually not know/difficult to know or with high local variability, then necessarily requesting some kind of averaging or simplification process in order to make the model actually applicable at the river basin/ reach scale.**

- To better show the possibility of parameterizing the **floodam.agri** model at different scales and how it has been calibrated for the development of national functions, we have taken and merged the former figures 3 and 4. We have highlighted the input data, intermediate outputs and final outputs in the new figure 3.

- In addition, the figure 8 has also been based on the figure 3, which makes it much clearer what is needed in terms of transfer, such as adapting to the resolution of hydraulic models, calibrating damage functions using local data, etc.

- **GC 9.1: how the problem of crop rotations was handled for identifying crop type in each plot?**

  – Following the suggestion of referee 1, we have added this clarification in section 3.2: "For the national application, we proposed to create a mixed function to manage rotations if necessary on the application territory. If the 3-year rotation is wheat, wheat, rape, the weight assigned to the wheat function is 2/3 and the weight assigned to rape 1/3. This explanation will be added in the section case study/ready to use flood damage functions."

- **GC 9.2: In which month of the year the flood was supposed to occur in the damage calculations (did they consider the month with the highest probability of flood occurrence / or did they calculate a weighted average damage, with the weights represented by the probabilities of flood occurrence in each specific month ?)**

  – We have better explained in the new version (section 3.2 and section 4.4) the difference between the time step used to model damages (week) and the groupings that were made to produce the national damage functions.

**General comment 10: I finally do not agree with the statement at P27.L615-616 (as well as the one in the abstract, P1.L5-8), or I may did not interpret well the Authors' point, which I then ask to be better clarified. According to me, the model framework must be certainly general, but, especially when modelling damage to agriculture, it is strictly necessary to be context- specific, in order to capture the typical features of the region where the model is applied, otherwise we are oversimplifying reality, which can be an acceptable solution (a "full adaptation" may be a huge effort, since while some components can be easily adapted (e.g. yields and price values), there are others that highly change from a context to another (e.g. the cultivation practices and operations)), but that we must be aware of it**

- This important remark of referee 1 has been taken into account at two levels:
  – First, in the new version of the article, the new figure 3 highlights much more clearly the generic and specific parts of the model. The inputs needed to feed the model are more clearly described. Secondly, the Transferability section shows clearly from the figure 8 how these inputs should be adapted according to the transfer application study case.

– Secondly, we have added section 5.5 which presents the perspectives for improving the process-based model for agricultural damage assessment. In particular, it shows the challenges that remain to be addressed in terms of taking into account adaptation behaviors at the individual scale and the need to consolidate the observation part.

**Minor comment: although the sense is always clear and English usage is almost correct, the paper needs to be proofread as there many typos (e.g. "litterature", "diven", "developped", just to cite some) and few weird sentences (e.g. P2.L24, P2.L41).**

- We have carefully proofread the document, correcting obvious typos. For the verification of the turns of phrase, we will rely on the NHESS proofreading.

**2 Summary of changes regarding observations and recommendations from referee 2**

We would like to thank referee 2 for the analytical work that was done on our paper. We thank him for his encouragement to publish this work and for all the suggestions and questions that were formulated and which helped us to improve this paper.

In the following, we have numbered the general comments from 1 to 5. Sometimes comments have been subdivided into sub-sections in order to respond specifically to each suggestion. This is the case for the general comment 3. Then specific comments (SC) are numbered from 1 to 6. One technical comment seemed to us to require a more detailed response and we have added it to the specific comments. This is the SC 6. Referee 2's comment is given in bold and our response follows the comment in plain text. All technical comments have be taken into account without the need for a specific response.

**General comment 1 : The authors propose a methodological framework to understand under which conditions expert knowledge used to fed process-based models of flood damage assessment are valid. Their framework is based on 4 axes: explication of assumptions, validation, updatability and transferability. an application is proposed in France for the agricultural sector. The focus given to the agricultural sector is well justified by the fact that agricultural lands are often flooded to reduce urban flood risk. Assessing flood damage is thus key to evaluate the efficiency of this measure and the compensation given to farmers. This article is very valuable contribution because it proposes a framework for flood damage assessment which is generalizable and it claims to make explicit the assumptions used in such models. Furthermore, it proposes an open source model for flood damage assessment in agriculture in the form of a R package, to be available soon.**

- We thank referee 2 who encourages us to publish this work which seems to us important to go towards a mutualisation and a capitalisation of the modelling effort to better evaluate the impacts of floods.

**General comment 2: The model is applied to the agricultural sector. It is restricted to the plant farming. What about livestock? Is this also applicable to this sector of agriculture? It could be discussed**

- We have taken this remark into account on two levels:
  – First, in section 4.1, we added the cattle component to the figure 5 and indicated in the text that this component was not currently supported in **floodam.agri**.
  – Secondly, we have added the section 5.5 on potential improvements to be implemented for the assessment of damages to agriculture. In particular, we propose ways to improve the consideration of damages to livestock by considering the work that has been done by the FHRC.

**General Comment 3: on taking into account the farmer's decisions in the model**

- **GC 3.1: When I look at your system of decision I cannot see a symmetry between the crop and plant material systems. You include the possibility to change the crop type in**

**equation 8 but not for plant materials. One should also have the case of a farmer who decides to plant another type of trees, similarly to equation 8 for crops.**

– This comment from referree 2 allowed us to clarify the issue of decision modeling in several way:
   * First, in section 4.1, we have explained the assumption of continuation of the current activity and reconstruction used for all components to produce the current damage functions with **floodam.agri**.
   * Second, in section 4.1 part EA3 dealing with decisions, we made several changes:
      · the modeling of the behavior in standard situation (crop management sequence) and its adaptation in case of flooding have been better described with the figure 6 and the table 6.
      · we have summarized in the tables (7 and 8) the decisions related to plant material and crops, illustrating with the case of apple cultivation.

   * Third, the "decisions related to crops" section has been expanded to better explain the strategies for reseeding or planting a spring crop. The table 8 also makes it clearer to which crop types the strategies apply.

   * Fourth, section 5.5 has been added to take into account possible improvements and, in particular, long-term adaptations such as the conversion of perennial crops (viticulture, arboriculture) to annual crops or grasslands.

- **GC 3.2: It seems to me that not all the post flood decisions made by farmers should be taken into account in the model otherwise you overestimate the damage. This is particularly the case when farmers decide to do something different from what they were doing before the flood (like in equations 8 and 12, sowing another crop or not replanting). In this case, the variation of revenues is not a damage because the reference has changed. The pre and post yields are not comparable, Y\_new is different than Y\_u because it is another crop, not because the biophysical conditions have changed in the farm because of the flood. If a farmer decides not to replant trees or crop, for example because he/she stops the activity or because she/he wants to invest in another farming activity or other species for example, then the damage function (eq 8, eq 12) is rather an opportunity cost or possibly a benefit rather than a damage. Counting equation 8 and 12 as a damage creates opportunities for farmers to operate a change in their agroecosystem and ask for money to the damage compensation organism for that change because they have been flooded. But the reason is not the flood, the reason can be economic or another reason. This will also have the perverse effect of making farmers prefer to wait to be flooded to change their agroecosystem to receive more money (in the case where they are compensated based on your damage functions.) This does not mean that the famer cannot anymore change the crop system after a flood, but it means that the compensation based on the damage function should not pay for the change but pay for what has been lost. To pay for the change brings your model to the context of adaption to climate change, not a context of compensation for flood damage. One could imagine a farmer willing change species in order to use species more resilient for floods because floods become more frequent or more devastating. This is possible but this is not what your paper is proposing. Your paper is about compensation, not adaptation. This should be discussed or corrected.**

   – Taking into account the previous suggestion, we have also responded to this remark.
      * the rules for replanting decisions have been specified and illustrated with the case of apples (table 7).
      * the strategies for reseeding or planting a spring crop for annuals have been better described in the text and in Table 8. We show that in the case of sowing a new crop the product cannot be higher because of the loss of cultivation costs incurred for the previous crop. We have also specified that this practice is only possible under certain conditions (season, integration in a rotation).

**General comment 4: Section 4.2 validation: V2 on comparability with other models (uk , Italy,**

etc). **Maybe you can compare the conceptual approaches between UK, Italy and France. This can help you to also highlight the contributions of your model to the literature. By literature I do not mean the case study based literature (filling the gap of having a model for the French agriculture) but the literature on the structure of flood damage assessment models (ie your figure 8). For example, is it usual to integrate decision rules in the calculation of damage or the biophysical processes? This kind of comparison will improve your contributions (in addition to the contribution of making explicit the assumptions) and the value of the paper for an international readership.**

- As proposed, we have attempted to make a comparison of the process-based models identified for agricultural damage assessment (Table D.1). We used the methodological framework proposed in the article as a basis for comparison. However, based on the existing literature on these models, a certain amount of information is still lacking and should be consolidated in a future work.

**General comment 5: I recommend to have the paper revised by a native English speaker: grammar, use of the article "the" (the figure x , the table x versus Table x, Figure x), etc.**

- We have corrected the English in the article as best we can, taking into account the suggestions of the two referees. We will then rely on NHESS proof-reading to further improve the quality of the language.

**Specific comments**

- **SC1: Tables are at the end of the paper (except Table 1) and figures in the main text. Are the tables part of an appendix or to be included in main text? If they have to be part of an appendix, please check the guidelines for authors.**

  - This technical problem has been resolved in the new version. The tables that should be included in the appendices have been clearly identified.

- **SC2: Plant material or perennial crops? You have related plant material to perennial crops line 318 but you have an equation for perennial crops in the section related to crops and then several equations in a section on plant material. This is confusing.**

  - This comment from referee 2 invited us to review the order of the sections on decisions. Crop losses are dependent on plant material losses and replanting strategies for the portion of the orchard or vineyard where there are plant material losses. We therefore placed the section on plant material decisions before the section on crop decisions. Crop losses related to plant material are better explained in this section (Table 7 and accompanying text).

- **SC3: Equation 8: What happens if Y_new > Y_u? It is no more a damage but a benefit. Does this mean that the farmer will revert money to the compensation fund because she/he earns money after the flood? This should be discussed or a constraint should appear in the system of equations**

  - The strategies for reseeding or planting a spring crop for annuals have been better described in the text and in the table 8. We show that in the case of sowing a new crop the product cannot be higher because of the loss of cultivation costs incurred for the previous crop. We have also specified that this practice is only possible under certain conditions (season, integration in a rotation).

- **SC4: Section Decision related to soil. It seems to me that you should also discuss the case of a variation in soil quality because of the flood (example of chemical pollution, or loss oforganic matter of the first layer of the soil). This affects yield also. Does this correspond to equation 6? Or would this be a case of double counting if you add an equation for that?**

  - This remark has been taken into account in section 4.1, EA2: biophysical processes, in the part dealing with soils. In addition, an illustrative table of soil damage considered for arboriculture has been added (Table 8).

- **SC5: Figure 8. Following my concern about accounting for decision rules and actions in the modelling of the damage functions. My concern is now visual: depending on the decision/action, the farmers has the possibility to increase the damage if he/she chooses the appropriate action. To maximise the damage and so the future compensation can become a strategy for the farmers in this model. This is a perverse strategy in my sense but your model allows it if I understand it well. The damage should be based on past losses not on future losses in case of changing practices. I am Ok with accounting for future losses in case of deterioration of soil quality, or in case of sowing the same crop again**

    – We think that the way decisions have been modeled in **floodam.agri** is more clearly described in this new version (see previous answers) and that it avoids possible confusions on the issue of strategy changes.

- **SC6 : Farm building. Is the cadastre a possible source for the data on agricultural buildings? What are the limitations to not use it if it exists?**

    – In response to this question, we have added section 5.5 with potentials for improvement and in particular a section dealing with agricultural buildings.

**Technical corrections**

- We thank referee 2 for his technical corrections that have been taken into account.

---

## Author Response (AR2)

**Process-based flood damage modelling relying on expert knowledge: a methodological contribution applied to agricultural sector**

Point by point answer of the authors to referees comments for minor revision

Pauline Brémond et al

2022-09-02

**1 Summary of changes regarding observations and recommendations from referee 1**

We thank again the referee 1 for his second review. We are pleased to have been able to respond to its main criticisms.

**Suggestion 1: The Authors have specified (L145 and 235) that the model "validation" has been carried out with the same experts who were interviewed in the development phase of the model. I am still convinced that this process cannot be called "validation", given that, to be so, it would require (at least) different experts involved in the two different phases. The approach followed by the Authors may be fine (as I am aware that it could be difficult to find many experts available to be interviewed), but I would rather define this process as "sanity check of the model" or just "check of the model".**

- Referee 1's suggestion shows us that we had not insisted enough on the double interest of validation workshops with experts. Indeed, on the one hand, the coherence between the elements collected in the individual interviews and the implementation of the model is ensured (what referee 1 calls check of the model). Beyond that, the interest of these workshops is to allow the experts to readjust their assumptions according to the collective discussion, but above all to allow them to readjust their assumptions by presenting them with the entire modelling chain (loss of plant material, yield, associated behaviours)
- The section on validation was reworded as follows : "The aim of these workshops is multiple. They allow the coherence of the information collected in individual interviews to be verified and discussed collectively. Above all, they allow the results of the overall modelling chain (loss of plant material, yield, associated behaviours) to be presented to the experts who were interviewed separately on the different components of the model and to allow them to readjust their assumptions if necessary."
- We agree that involving external experts would consolidate the validation and we reworded the section discussion on validation as follows: "Two avenues are usually identified: first, the comparison of model results with each other; second, the comparison with claims data (Molinari, 2019c). A third avenue is to consider the geographical transfer of models as an opportunity to capitalise on expert knowledge by involving new experts and being able to clearly present the modelling assumptions to them."

**Suggestion 2: Always regarding Axis 2 ("Validation" in Table 2), points V3 and V4 pertain to model "usability"; then, I would suggest renaming the title for this Axis as "Validation and usability". Here the term "validation" is ok, as it is used in a more general context (including traditional validation against observed data).**

- We agree that part of the validation presented in axis 2 is related to the operational aspects of the usability of the model. However, we believe that changing the title of the section could lead to confusion

and we propose the following change in the section framework / validation: "The third modality is a validation related to the operationality and use of the model."

**Suggestion 3: I suggest the Authors to rephrase and better explain the following sentence "In this paper, we argue that process-based models are not doomed to be context specific as far as the modelling process is rigorous", both in the abstract and in the conclusion section. In my opinion, as it is now, it could be misleading, given that process-based models are always general in their methodological framework, while it is only their "application" that it is context-specific (as it also emerges from Authors' paper). I suggest here a possible amendment for the mentioned sentence: "In this paper, we argue that process-based models, based on a rigorous modelling process, can be suitable to be applied in different contexts.**

- We thank the referee 1 for his suggestion of amendment. We reworded the sentence as suggested.

**Description of damage to equipment: I guess Equation 16 implicitly implies some damage ratio (or vulnerability function) or are you assuming that equipment is always fully damaged whenever there is a flood? Please clarify this point.**

**Minor comments**

- L2: "because they may have greater exposure and are complex economic systems": in my opinion, it would be better to say: "because they are complex economic systems particularly exposed to floods".
    - the proposition has been accepted in the text
- L14-15: not so clear sentence; please consider rephrasing (perhaps by shortening it).
    - We have reworded it as follows : "We show in this paper that the proposed methodological framework allows an explicit description of the modelling assumptions and data used, which is necessary to consider a reuse in time or a transfer to another geographical area. In this sense, this methodological framework constitutes a solid basis for considering the validation, transfer, comparison and capitalisation of data collected around models based on processes relying on expert knowledge."
- L68: it seems there is a missing word after "insurance", probably "penetration" or "coverage".
    - this has been corrected
- L82: replace "it" after "(ii)" with "damage" to be more clear.
    - We changed "it" by "the loss of yield"
- L102-105: delete this paragraph, since it is just mentioned a few lines above (L93-94).
    - ok
- Please check figures and tables numbering: for example, I noted that Figure 3 is cited in the text before Figure 2 (the same applies to Tables 9 and 6).
    - This has been corrected.
- I would suggest merging Figures 3 and 8 into just one figure.
    - As the modelling process and its steps is quite complex, I am afraid that putting the modelling steps and the transfer steps on the same figure will lead to confusion. I prefer to keep the current form.
- L205-206: unclear sentence; please consider rephrasing.
    - We reworded as follows:"floodam.agri includes generic parts and can produce damage functions at different scales, depending on the calibration. We illustrate in this article the use of floodam.agri to produce damage functions at the national scale"
- L273: "materiel" -> "material".
    - ok
- Table 2: this information could be directly provided in the text (e.g. at L280), without using a table.
    - done
- L305: sub-questions are actually two and not three.
    - ok
- L319: insert a comma after "material" and "mortality".
    - ok
- Parameter beta (as in Table 7): I guess there are different thresholds for beta (which drives the farmer's

strategy) for different types of plants, or, based on expert knowledge, are they fixed values? Please specify this point.

- We reworded the paragraphconcernaning plant material to make it more explicit. beta is not fixed as presented inthe diagrams of the table 3. It is a function of water depths, flood duration, and velocity. I specified also that the tresholds given inthe ex table 7 (now table 6) are the assumptions made for the application at national scale.

- L467 and 469: "data" instead of "date"
  - ok

**2 Summary of changes regarding observations and recommendations from referee 2**

We thank again referee 2 for for his second review. We appreciate his encouragement to publish this article and are pleased to have been able to respond to his recommandations and questions in the first round.

**Suggestion 1 : On validation L409-416. you recommend (Axis V1) to compare estimation with sinistrality data, by comparing the output (as you say line 137). I guess by output you mean the final damage in monetary terms. But output is only one element of the comparison. The difference in the assessment method is another element. As you say L 413-416, your model uses more type of expenses after flooding than the compensation systems. So, why are sinistrality data considered as a reference data to compare your model with? Sinistrality data are also subject to assumptions, limitations. This issue is crucial for Cost-Benefit analysis because depending on the model used, a project can be approved or rejected. But both methods are valid as long as the assumptions are accepted by society (V3 on stakeholder expectations).**

- We fully agree with referee 2 concerning this aspect. To make this more explicit we have added the following sentence to the section framework / validation : "In addition, sinsitrality data should be considered with caution as it may only represent part of the damage that one wishes to compare. The insurance coverage of the different types of damage, in particular in agriculture, is not complete."

**Suggestion 2: On transferability. Can we think that your model could be used by a system of flood damage compensation where damage are not collected (because it is too costly for example) but estimated with your model? I think this would deserve to be discussed at some point in the paper or you should define the limit of transferability.**

- We thank the referee 2 for this suggestion. This raises an issue that we had not discussed so far in the article, which is to use the model for post-flood damage estimation from partial data. To take into account this interresting suggestions, we reworded the section discussion/consolidate validation as follows: "The proposed framework allows for a clear improvement in the validation methodology with experts involved in the modelling process. However, we are aware of the need to consolidate this aspect. Two avenues are usually identified: first, the comparison of model results with each other; second, the comparison with claims data (MolinariD2019c). A third avenue is to consider the geographical transfer of models as an opportunity to capitalise on expert knowledge by involving new experts and being able to clearly present the modelling assumptions to them.We consider that the clarification of the assumptions is a prerequisite for both avenues and the framework presented here is a step towards the possibility of comparing models with each other. We have made a first proposal in the table C1 based on existing literature. This should not be considered as a result but as a discussion support to allow exchanges on methods with a view to capitalization. Concerning the collection of ex post damage data, in particular for the agricultural sector, this is a real challenge that requires a long-term effort. Some interesting initiatives are to be highlighted, as for example, the validation carried out by (ChauVN2015) or (ShresthaBB2021a). The modelling effort we have carried out to develop floodam.agri has highlighted the importance of acquiring knowledge both on biophysical and human processes in order to be able to assess damage in economic terms. This implies that the data to be collected post-flood in order to validate a model such as floodam.agri must be of different natures, ranging from biophysical impacts (yield loss, mortality of plant material, soil erosion...) to monetary damage, including the chain of

behaviours of recovery and continuation of crop management sequence. But this type of post-flood data collection is very time consuming. Most of the time, on large-scale events, the primary objective will be to obtain an overall damage assessment fairly quickly and not to carry out a detailed characterisation of the damage formation processes. In this case, it could be used to estimate damage in monetary terms from hazard parameters. It could also be used to estimate damage in monetary terms from partial post-flood data collection such as yield losses, which corresponds to the practice of the insurance system in France. This type of use would provide a more complete picture of the damage on the basis of the current modelling assumptions, but would not or only partially validate the estimated values. On the contrary, such a characterisation makes sense for small-scale events for which, however, various levels of impact can be encountered on an individual scale. In this case, the collection of data allows for validation. For this, the implementation of observatories is an interesting approach."

**Technical corrections/clarification**

The following corrections have been done.

- Line 182: What do you mean by local communities? Do you mean municipalities?
  – communities has been changed by locl flood risk managers
- Line 373: delete "are"
- Line 483: "the" week.